# Research on the generative logic and configuration effects of the policy implementation environment in China's grassroots digital construction: Traceability based on grounded theory and the validation of the csQCA method

Junjie Li[1,2☯]*, Bangfan Liu[1,2☯]*

1 School of Public Administration, Yanshan University, Qinhuangdao, China, 2 Hebei Public Policy Evaluation and Research Center, Qinhuangdao, China

☯ These authors contributed equally to this work.
* lijunjie@stumail.ysu.edu.cn (JL); liubangfan@ysu.edu.cn (BL)

**Data Availability Statement:** All data comes from China National Knowledge Infrastructure (https://www.cnki.net/) And WOS database (http://

## Abstract

Using NVivo12plus software, this study constructs a generation model of the policy implementation environment in China's grassroots digital construction by examining the data from interviews with 37 Chinese grassroots civil servants as its research object. At the same time, with the help of the csQCA method and on the basis of rooted coding, using Tosmana software, the model validation and model expansion of 37 cases were carried out. This study shows that the main components of the policy implementation environment in China's grassroots digital construction include five main dimensions: cognitive, emotional, behavioral, normative and control. At the initial stage of China's grassroots digital construction, the cognitive environment dimension is the most critical link in the environmental governance of policy implementation. The cognitive and normative environments dominate the effect pattern of China's grassroots digital policy implementation. There are six types of motivation patterns for the environmental effect of the implementation of grassroots digital policy: know-doing-norm, cognitive-dominant, know-doing-control, emotion-control, behavior-lacking and regulation-dominant types. Based on the above analysis, there are three main policy suggestions to promote the generation of a productive policy implementation environment and positive policy effects in China's digital construction: "degree", "force" and "sense". These suggestions include strengthening the top-level design, improving the integration and cooperation degree of the environment, taking the root at the grassroots level to improve the adaptability of the environment, as well as increasing publicity and promotion to improve the sense of environmental experience.

webofscience.com/) Readers can freely obtain the data used in this article based on the search criteria designed in this article.

**Funding:** The authors received no specific funding for this work.

**Competing interests:** The authors have declared that no competing interests exist.

# 1. Introduction

Digital construction at the grassroots level is a significant element of grassroots governance and an important support for the modernization of the grassroots governance system and capacity. On the one hand, grassroots digital construction is a process of technical empowerment for the modernization of the grassroots governance system and grassroots governance capacity. On the other hand, the modernization of the grassroots governance system and grassroots governance capacity is the management empowerment process of grassroots digital construction. The policy environment is also an important form and carrier of this process of management empowerment. The so-called policy implementation environment refers to the sum of all the conditions that affect the implementation of the policy. The implementation of the policy requires the implementation of the policy; therefore, the policy implementation environment is the most critical part of the policy environment's effectiveness.

In recent years, the Chinese government has attached great importance to grassroots digital governance and has achieved certain governance results. Some research on grassroots digital governance has also been carried out by the academic community. Fang Fei et al. consider that instrumental rationality over value rationality is an important reason for the practical difficulties of China's targeted poverty alleviation policy and that it is necessary to weaken instrumental rationality and strengthen value rationality in future poverty alleviation governance. Yuan Mingbao considers that digitalization and textualization have brought about a lack of poverty governance [1]. Wang Yaling posits that the lack of public participation leads to problems in smart city construction, such as focusing on technology over application, stereotyping, insufficient public perception, and the digital divide between urban and rural areas [2]. Huang Jianwei et al. state that while it is important to solve the technical dilemma of grassroots digital governance, it is more important to pay attention to the construction of ethics and promote the humanization of digital governance [3]. Sun Zongfeng et al. consider that technology, avoidance of responsibility and field logic constitute the triple logic of the implementation of targeted poverty alleviation policies at the grassroots level, among which field logic has always existed in the implementation of grassroots policies [4]. Yu Haichun considers that there are three main ways to improve the implementation of the system: one is to optimize the allocation of resources at the grassroots level, the other is to improve the performance appraisal mechanism, and the third is to improve the adaptability of the bureaucratic system [5]. Han Ruibo argues that the construction of digital countryside needs to establish a balance mechanism between technical governance and autonomous governance. Among them, technological governance is dominated by administrative forces, while autonomous governance is dominated by social forces [6]. Chen Xin et al. consider that the distortion of superior attention is the main cause of digital formalism. To break digital formalism, it is necessary to build a three-dimensional governance model of the synergistic evolution of "value–institution–technology", run through public value in digital transformation, give full play to the advantages of digital technology in flexibility and mobility, and make bureaucracy more suitable. The public value should be integrated into the digital transformation of the government, and the flexibility and mobility of digital technology should be brought into play to improve the adaptability of bureaucracy [7]. Fu Liping et al. consider that township cadres are mainly influenced in two ways to carry out technological governance actions; one is to increase accountability for policy implementation, and the other is to use technical means to compress the discretionary power. After identifying the characteristics of policies, township cadres adopt strategies to carry out power rent-seeking, formalism, avoidance of responsibility and procedural activities [8]. Wang Xiaoyan posits that the current internal entanglements of e-government include information distortion, implementation deviation and correction difficulties. She notes that digital

governance is the transformation direction of e-government, which needs to realize three changes: first, from the technology-embedded system to technology-embedded governance; second, from the affairs center to the personnel center; and third, from input resources to strengthen innovation and reform [9]. The rapid development of artificial digital technology in Back East (2021) presents both opportunities and challenges to policy issuance. Based on historical experience and national conditions, it is believed that the digital transformation of policy issuance should be gradually promoted and implemented at different levels [10]. Huang Liuzhao considers that the deviation of grassroots policy implementation is the normal state of grassroots governance, which has reached a consensus in the academic circle. From the perspective of policy design itself and the mutual construction of interest subjects, digital competition and double game are one explanation for the implementation deviation of poverty alleviation and relocation policy in inhospitable areas [11]. Tong Linjie et al. consider that digital formalism is an evolving form of formalism in the information age. It is the result of the combination of digital forms' substitution of digital content, rational choice of policy implementation deviation, longitudinal pressure transmission among bureaucracies, lack of multiple reward and punishment mechanisms and alienation of accountability [12]. Jiang Bao et al. note that the decoupling of government decision-making and government execution is the main path to compatible advantages of the hierarchical organization model and the flat organization model, and the reform of digital government organization needs re-coupling among organizations on the basis of decoupling [13]. According to Li Xiaofang et al., the types of digital formalism that should be given special attention include controversial, execution distortion, "cater-display" and redundancy types [14]. Shao Chunxia et al. argue that digital technology interacts and influences governance actors. The bureaucratic system's own digital needs as well as the willingness and utilization of the bureaucratic personnel will restrict the reconstruction of the bureaucratic system using digital technology [15]. Dong Shitao et al. consider that the technology execution system should be reformed to promote the positive interaction between technology and organization so as to correct the formalism of digital governance [16]. Xu Chang et al. consider that system design and community mobilization are the decisive factors in the formation of the collaborative governance pattern of grassroots social mobilization. Among them, system design plays a fundamental role, community mobilization plays a guarantor role, and trust and digital technology play a positive regulating role [17]. Yang Weimin posits that digital formalism is one of the forms of formalism in grassroots governance and that the regulation of grassroots formalism requires the empowerment and empowerment of information technology [18]. Yao Qingchen et al. consider that ensuring the data supply relies on system, mechanism and technological breakthroughs; efforts should be made in concepts, policies and actions to correct governance deviations, and a solid foundation should be laid in literacy publicity and legal aspects to deal with cyber risks [19]. Ding Yizhou considers that the key to transforming institutional advantages into governance effectiveness is to compare the implementation of policies before and after digital empowerment to find out the advantages and disadvantages of digital empowerment [20]. Yu Lian et al. consider that there are three main mechanisms for digital assistance in helping county governments implement policies: the power concentration mechanism in the information transmission link, the coercive supervision mechanism in implementation and the promotion incentive mechanism in assessment. He pointed out that digital technology helps the establishment of coordinated county governments, but it is also necessary to be wary of digital distortion, digital prison and digital leviathan brought by excessive digital dependence [21]. Lu Jiangyang et al. note that digital empowerment is one of the main ways to correct the deviation of governance behavior of grassroots governments [22]. Fan Weifeng et al. argue that the deep-seated reasons for digital levying mainly include four aspects: firstly, digital governance caused by digital cognitive bias

is divorced from the actual situation at the grassroots level; secondly, there is excessive embedd-edness in the integration process of bureaucratic governance and digital governance; thirdly, there is internal tension between social governance and digital governance; and, fourthly, a "technology execution trap" is formed when digital technology is embedded in grassroots governance practices [23]. Dong Youhong et al. consider that to analyze the generation mechanism and deep logic of the overload of grassroots urban transport platforms, it is theoretically necessary to use the "situation-goal-structure" analysis framework to explore the strategies and paths to resolve the overload under the guidance of this theoretical framework [24]. In addition, on the one hand, it is necessary to strengthen the overall coordination ability of grassroots platforms, and, on the other hand, it is necessary to give full play to the technical enabling advantages of grassroots urban transport platforms and promote the digital transformation of urban governance to achieve improved results at the grassroots level of streets and towns.

To sum up, the current research on grassroots digitization policy implementation in Chinese academic circles mainly focuses on the ontology of grassroots digitization policy implementation, that is, the analysis of the technical, organizational, institutional or specific policy implementation difficulties encountered in grassroots digitization policy implementation, and the lack of grassroots digitization policy implementation as a systematic project. From its internal and external environment, the internal mechanism of its generation and the interaction between its internal and external environment are considered comprehensively. Policy implementation in grassroots digital construction is not only affected by the policy itself but also restricted by the implementation environment, which is dialectical unity. Therefore, the aim of this study is to examine and verify the policy implementation environment of China's grassroots digital construction from the perspective of system and environment. Specifically, it mainly focuses on four major research questions: one is the generation logic of the policy implementation environment in the grassroots digital construction; the other is whether the policy implementation environment in the grassroots digital construction is conducive to policy implementation, that is, the evaluation of the implementation ability of the grassroots digital policy environment; the third is the multi-factor combination situation and explanation of the cause of the implementation ability of the grassroots digital policy environment. The fourth is the specific path and countermeasures for the optimization of implementing the grassroots digital policy environment. It focuses on the configuration analysis and verification of the generation logic of the policy implementation environment in grassroots digital construction and realizes a certain degree of theoretical and practical innovation. In terms of theoretical innovation, the first aspect is to build a generation model of the policy implementation environment in grassroots digital construction. The second aspect involves explaining the internal mechanism of policy implementation environment generation in grassroots digital construction. Lastly, the third aspect entails building a configuration model of grassroots digital policy environment implementation. From a practical innovation perspective, the countermeasures and suggestions for optimizing the implementation environment of grassroots digital policy are put forward to provide an important basis and reference for the practice of grassroots digital policy environment optimization.

## 2. Research methods and data sources

### 2.1 Research methods

Grounded theory is a qualitative research method. Qualitative research is "an activity that integrates the researcher into the research environment, uses multiple methods to collect data, studies social phenomena, and analyzes the data by induction in order to form theories and obtain interpretative understanding." Grounded theory is considered to be the most scientific

research method in qualitative research. It adopts qualitative methods in research design and data collection, and quantitative methods in data analysis [25]. This study chooses the analysis paradigm of grounded theory based on the following two points: first, the macro- and micro-environments that affect the implementation of grassroots digital policies are constantly developing and changing. At present, the theoretical circle has not reached a unified and mature evaluation index system and statistical scale, which needs to be further explored; second, the practice of grassroots digital governance is restricted by a variety of subjective and objective factors, and the heterogeneity of social conditions, people's feelings, ideas and other aspects is large, so it cannot copy the ready-made theories, methods and models at home and abroad. Data collection and coding is the core step of rooted theory. The coding process is mainly divided into open coding, spindle coding and selective coding, and research conclusions are drawn after theoretical sampling.

This study extracts the generative logic of policy implementation environment in grassroots digitization construction through rooted theory. The specific research vein mainly includes four parts: the first is data collection. Through in-depth interviews, focus group interviews, participant observation and other methods to obtain the original interview data, the research portrait is drawn. The second is theoretical analysis and model construction. Through open coding, the data are comprehensively combed; through spindle coding, the data category structure is formed; by choosing coding, we can find an accurate story line for the research problem and construct the theoretical model. The third is model interpretation. Inductive logic, combined with research problems, is used to further explain the model systematically. The fourth is the conclusion and enlightenment. Based on the above analysis and research, this study puts forward targeted countermeasures and suggestions on the governance dilemmas existing in the policy implementation environment in the construction of grassroots digitalization.

Qualitative comparative analysis (QCA) is a kind of configuration comparative analysis method. This method regards positive social phenomena as the different results caused by the complex combination of different attribute factors. Qualitative comparative analysis, or QCA, has three main technical forms: CsQCA, MvQCA, and FsQCA. CsQCA is a QCA technique developed in the late 1980s by Charles Ragin and programmer Kriss Drass [26]. It is by far the most widely used QCA technology. Its main advantages are reflected in four aspects: first, it conforms to the principle of simplicity. CsQCA is the most widely used technology in the political science field among the three QCA methods. Based on a Boolean set, CSQCA can simplify complex phenomena while better protecting potential interests and related phenomena. Second, it is not limited by the sample size, and the robustness of the result lies in the representativeness of the sample. Third, it is suitable for explaining multiple complex causality but not linear causality. Fourth, qualitative research is often questioned for its lack of universality and loose structure, while the essence of QCA iteration supports a moderate degree of universality, can realize the systematic comparison of complex cases, can be transformed into logic, and has replicability, thus providing QCA with "scientific" characteristics and reducing the fuzziness and subjectivity of traditional qualitative research [27]. In summary, first, the policy implementation environment cases in the grassroots digital construction are not the usual digital data, and it is of little significance to set an accurate threshold; second, there are 37 research samples and 5 explanatory variables in this study, which meets the QCA's recommendation of generally selecting 4–6 interpretation conditions and 10–40 cases; third, the policy implementation environment in grassroots digitization construction involves many aspects including organization, technology, system and mechanism and is affected and restricted by compound causality, which is not suitable for linear causality explanation. Therefore, CsQCA is more in line with the attributes of the research object in this study and is the most suitable QCA technology for this study.

## 2.2 Data collection and sources

In the process of collecting research cases for this study, MPA students from the class of 2022 in Yanshan University were interviewed about the topic of this study. The interviewees were clearly informed before the interview; that is, the content of the interviews forms the research case of this study, which only involves the views pertaining to the topic of this study to avoid breaching personal privacy. All interviews were approved by the interviewees, and the whole interview process was recorded. All interviews were with adults. The interviews for this article were conducted from May 10 to June 15 2023.

Through semi-structured "face-to-face" interviews, realistic materials were collected on the generative logic and execution dimensions of the policy implementation environment in China's grassroots digital construction. In the interview process, open questions were asked, such as "How do you evaluate the digital policy environment in your organization?", "What factors influence your implementation of the digital policy?", etc., to obtain the respondents' generative logic and execution information about the policy implementation environment. A detailed outline of the interviews is shown in Table 1.

In order to collect samples, the researchers conducted interviews with a total of 40 grass-roots civil servants through Tencent conferences, WeChat voice calls, face-to-face interviews and other forms of communication from April to May 2023, and recorded the interviewees' information (see Table 2). The duration of the interviews ranged from 4 to 25 minutes due to differences in the extent of interviewees' enthusiasm for the topics of interest. Interviews were recorded and saved onto a computer by the researcher at the end of the interview. In view of the needs of the interview, a combined group interview was conducted with some interviewees from the same unit. Therefore, a total of 37 electronic documents were formed, which became the main object of rooted theory coding. During the coding, NVivo12plus and ROSTCM were used to assist the analysis. Thirty-four interview documents were selected for coding analysis and model construction, and the remaining three were used for saturation testing of rooted theory.

**Table 1. Interview theme and content outline.**

| Topics of the Interview | An outline of the interview |
| --- | --- |
| Basic Information | Unit, title, gender, age, previous employment, and whether or not you are involved in digital work |
| Awareness of the policy implementation environment in grassroots digital construction | (1) What are the policies on digitalization and digital government involved in your work? How do you understand these policies?<br>(2) What digital platforms are involved in your work? What is the development status of these digital platforms? Has digitalization solved social problems that are difficult to be effectively solved by traditional governance means, especially the "pain points" and "difficult points" in social problems?<br>(3) How do you feel when implementing the above policies? What do you think caused you some pressure when implementing the above policy? The policy itself? Leaders assigning tasks? Being tied to performance reviews? Lack of competence? Information or data security? Confidentiality?. . . What difficulties have you encountered in implementing the grassroots digital policy? From your own perspective, what factors do you think will affect the implementation of the grassroots digital policy?<br>(4) If you were asked to evaluate the implementation effect of the above policies, what would you say?<br>(5) How do you think the above policies should be improved?<br>(6) Would you like to add anything to the discussion at this point? |

**Table 2. Information of the grassroots civil servants interviewed.**

| Id. | Interview subjects | Units | Position | Gender | Job duties | Interview format |
|---|---|---|---|---|---|---|
| N01 | Sheen ** | W City A City Tax Bureau | First-class Administrative Law Enforcement Officer | Female | Administration (official documents, letters and visits, etc.) | Face to face |
| N02 | Wang * | W City F District Finance Bureau | First-class Section Member | Female | We implemented fiscal budgets and subsidies | Face to Face |
| N03 | Liu * | K Street Office, District H, City W | Grade I Section Officer | Female | Health services and administration | Face to face |
| N04 | Zhao * | W City C County W Subdistrict Office | Section I Officer | Female | General Affairs | Face to face |
| N05 | Meng ** | Office, Street F, District F, City W | Section I Officer | Female | General Affairs | Face to face |
| N06 | Wang * | W City F District Tax Bureau | Section CHIEF | Female | Discipline supervision | Face to face |
| N07 | Wu ** | W City W District Tax Bureau | First-class Administrative Law Enforcement Officer | Male | Information collection and management | Face to face |
| N08 | Lee ** | W City, C County, Street B Office | Section STAFF | Female | Rural finance | Face to face |
| N09 | Liu ** | W District Bureau of Commerce, W City | Intermediate Skilled Economist | Male | Market order operation management | Face to face |
| N10 | Chang ** | W City W District Tax Bureau | First-class Administrative Law Enforcement Officer | Female | Assist, tip-off verification | Face to face |
| N11 | Song ** | Office of F Street, District F, City W | Section I Officer | Female | Financial management | Face to face |
| N12 | Lim ** | W City F District Tax Bureau | Deputy Chief | Male | Party building | Face to face |
| N13 | High ** | Baodu Sub-district Office, C County People's Government, W City | Business Cadre (Professional and Technical Personnel Intermediate Accountant) | Female | Economic management station teller | Face to face |
| N14 | Zhang ** | W City G District Tax Bureau | First-class Administrative Law Enforcement Officer | Male | Tax service | Face to face |
| N15 | Zang ** | W City K District Local Financial Supervision Bureau | Grade I Section Member | Male | Party affairs, office | Face to face |
| N16 | Liu * | W City G District Tax Bureau | First- class Administrative Officer | Female | Tax services | Face to face |
| N17 | Lv * | W City and L County Public Security Bureau | Deputy Squadron Leader | Female | Responsible for general affairs | Face to face |
| N18 | Hu ** | Q City and Q County Tax Bureau | Section Officer | Female | Tax refund, income forecast | Tencent meeting |
| N19 | Lee * | Q City C County D Town Government | Section Members | Female | Township propaganda | Tencent conference |
| N20 | Off ** | Commission for Discipline Inspection of C County, Q City | Section Member | Male | Discipline inspection | Tencent meeting |
| N21 | Hole ** | T City L Civil Bureau | Section Staff | Female | Elderly care, children with disabilities | Tencent conference |
| N22 | Wang ** | Q City and L County Development and Reform Bureau | Section Member | Female | Economic indicators assessment and calculation | Tencent conference |
| N23 | Lee * | Q City F District market supervision Administration | Section Member | Female | Credit and online transaction supervision | Tencent Conference |
| N24 | Wu ** | Q City and C County Bureau for Letters and Calls | Section Member | Female | Inspector-general | Tencent conference |
| N25 | Zhao ** | Q City C County Commissioners Office Inspection room | Section Member | Male | Co-ordinating | Tencent conference |
| N26 | Feng ** | Q City Public Security Bureau Beidaihe Branch office | Section Staff | Male | General administrative work | Tencent Conference |
| N27 | Chang ** | Q City B District Tourism and Culture Radio and Television Bureau | Section Member | Female | Comprehensive manuscript drafting | Tencent Conference |
| N28 | Xu ** | Q City B District Government Office | Section Member | Female | Document flow and handling | Tencent Conference |
| N29 | Beam * | T City International Tourism Island Management Committee Cultural Tourism Bureau | Section Member | Female | Publicity | Tencent conference |

(*Continued*)

**Table 2.** (Continued)

| Id. | Interview subjects | Units | Position | Gender | Job duties | Interview format |
|---|---|---|---|---|---|---|
| N30 | Liu ** | Q City H District Tax Bureau | Section Officer | Male | Export tax refund | Tencent conference |
| N31 | Pay ** | Q City C County Longjiadian Town People's Government | Section Member | Female | Poverty alleviation and prevention of return to poverty | Tencent Conference |
| N32 | Single ** | Urban Management Bureau office, District F, Q City | Section Member | Female | Party work, administrative affairs | Tencent meeting |
| N33 | Hole ** | District F Court, City of Q | Section Clerk | Male | Auxiliary judge for cases | Tencent Conference |
| N34 | Liu ** | T City and Z City League Municipal Committee | Secretary of the Communist Party of China League | Male | To organize and lead the work of the Youth League Municipal Committee | Tencent Conference |
| N35 | Liu * | T City Q City Shaheyi Town People's Government | Staff | Male | Petition to maintain stability | Tencent conference |
| N36 | Liu * | Q City B District Customs | Section Officer | Female | General office management jobs | Tencent Conference |
| N37 | Lee ** | Q City C County X Town Government | Section Member | Female | People's Congress, Youth League Committee, rural revitalization and poverty alleviation | Tencent Conference |
| N38 | Yu ** | District Court F, City of Q | Clerk | Female | Administration | Tencent Meetings |
| N39 | Lee ** | Q City F District L town government Party building office | Section Member | Female | Distance education, non-public party building, veteran cadres | Tencent conference |
| N40 | Shao * | Bohai Township, K District, Q City | Staff | Male | Community building, government services | Tencent Conference |

Here, two points need to be explained: first, interviews were carried out with the staff of the big data bureau. Although the big data administration (BDA) is an important government agency in China coordinating the construction and management of government information network systems, government data centers, e-government infrastructure, and government infrastructure and public informatization projects, given that it is a macro-planning and construction unit, the focus of this study is to examine the environment in which specific grass-roots digital policies are implemented, that is, the specific implementation of digital policies at the grassroots level. At present, China's big data bureau is only built to the provincial level, and no big data bureaus have been established below this level. As far as the current provincial construction is concerned, it is also the function of integrating the existing departments such as publicity, communication, information and data, so it can be predicted that the establishment of big data bureaus of governments below the provincial level will also adopt this principle. In this study, more staff members from the departments of publicity, communication, information and data were selected for this consideration. Second, in terms of the representation of selected groups, this study mainly interviewed MPA graduate students in Yanshan University, mainly based on two considerations: First, these MPA graduate students are working at the grassroots level, involving various departments and industries at the grassroots level, distributed in the Beijing–Tianjin–Hebei and Lu-Liao regions, with a considerable degree of representation. On the other hand, this study is a multi-case study of qualitative research. Different from sampling of quantitative research, it follows the basic rules of case study, focusing on depth rather than breadth. Through in-depth interviews, this study strives to dig and refine the information and content of respondents from different grassroots units to achieve longitudinal vitality. In view of achieve the reasoning effect from the case to the general theoretical results and providing support for the construction and verification of the following theory, the general

theoretical results have great value and significance in guiding the grassroots digital practice in China.

## 3. Rooted theory tracing

### 3.1 Rooted coding analysis

Interview document analysis processing is based on the data coding operation process and ideas determined by Glaser et al., including open coding, axial coding and selective coding [28]. In order to ensure the authenticity and reliability of the case study and avoid the deviation of human subjective understanding as much as possible, the text materials used for coding are composed of the original fragments of the collected materials.

**(1) Open coding.**　Open coding was used to analyze the interview text materials according to the conventional process of "original text materials–labeling–conceptualization–categorization", and formed 716 "labels" and 38 initial categories. Among them, each "label" corresponds to more than two original text statements. Each initial category lists one original statement as an example (Table 3).

**(2) Axial coding.**　The main task of axial coding is to extract the relationship between categories according to the coding mode. It is a complex induction and deduction process that links sub-categories and main categories [28], consisting of multiple steps. Through axial coding, this study further summarizes the above 38 categories into 16 sub-categories including self-cognition, external cognition, sense of security, sense of experience, sense of boundary, publicity and interpretation, coordination and integration, passive execution, autonomous participation, legal system, ideology, values, cultural customs, supervision and management, feedback reinforcement, and cost control and five main control environment dimensions including the cognitive dimension, emotional dimension, behavioral dimension, normative environment dimension, and control environment dimension. The above main categories and sub-categories of the relationship connotation are shown in Table 4.

**(3) Selective coding.**　The purpose of selective coding is to discover the core category from the main category and establish the relationship between the core category and other categories by way of story lines, so as to refine the theoretical model of research [28]. This is the final link of coding analysis and the formation of theoretical framework. The core category determined by this research is "the generative logic of policy implementation environment in China's grassroots digital construction", which consists of five main categories: the cognitive dimension, emotional dimension, behavioral dimension, normative dimension and control dimension.

The story line generated around this core category can be roughly understood as follows: the generative logic of the policy implementation environment in China's grassroots digital construction is a dynamic operation process driven by the cognitive environment dimension. The cognitive environment dimension of policy implementation is the basic cognition and understanding of the policy implementation subject to the situation of China's grassroots digital construction. As far as China is concerned, there will be differences in the individual cognition of various policy implementers in the process of implementing instrumental digital policies, but the fundamental cognition of policies is the same. It is precisely because of the consistency of the fundamental cognition of the policy that each policy implementation subject can take root in the unit and carry out the grassroots digital governance based on the position, which leads to the emergence of a series of policy implementation emotions and policy implementation behavior processes. To put it simply, the emotional dimension of policy implementation in China's grassroots digital construction is the comprehensive embodiment of the attitude of each policy implementation subject to their own environment formed on the basis

**Table 3. Open coding and examples.**

| Category (reference point/piece) | Original representative statement |
| --- | --- |
| Acknowledging Digital Manulife and the original intention of design (124) | Yes, I feel that the original intention of his setting up this system is to make the whole work more efficient, but in fact, when it comes to the grassroots level, the work is actually more tedious. (N02) |
| Digital coordination and integration (79) | It can't just be that there's no one to take charge of this thing, there's no way to share it, well each person does his own thing. (N04) |
| Cognition and quality of service object (52) | Implementation that is very important, there are some for example, um, because I stay in the district and county bureau, which belongs to the grassroots bureau, um, many taxpayers, are not highly educated, and um, before using paper invoices, they can go to the front desk to deal with, provide me with the billing information I want, the front desk will type out that information, um, now it becomes an electronic invoice, When the taxpayer needs to go online on his own, he can specifically operate on his own, and then, at this time, many taxpayers will say that I am old, and then I can't operate. (N01) |
| Digital products are not practical for grassroots work (45) | Well, I think that according to my current work, well, I think there are a lot of procedures on the system is too much patternization, well, there are some unnecessary procedures, but must go, that is, a simple thing is cumbersome. (N18) |
| Grassroots autonomy (45) | There is absolutely no such initiative to change the existing, he has to follow. (N03) |
| Supervision and control (40) | He will definitely have some omissions, so he said to supervise later. (N16) |
| Leadership perceptions and styles (35) | Yes, that is, our town in the e-government this area is not very developed, that is, no one to pay attention to this thing, including we go to the edge of the newspaper this information, as long as it is required to be two days a week, as long as it is the newspaper two days a week can, he will not ask too much about your channel, also will not be too concerned about your quality. (N19) |
| Data security and confidentiality (28) | Then the management committee should be our own organ party committee, they are responsible for themselves, because some work may be classified, it is not convenient to hand over to the operating company. (N29) |
| Allocation and transfer of attention (24) | No, it is placed on the side, because his data presentation also requires our grassroots staff to do a lot of right to do, but it is no longer needed. (N03) |
| Digital formalism (20) | Well, there's probably a lot of formalism here, too. (N15) |
| The reality is to increase the burden on the grassroots (19) | Well, if you say we go online, we may be, when we let the leader approve, even if we go online, we may also have to tell him face to face, we have gone online, please give us a batch, or call the leader to say. (N10) |
| System design (18) | Well, in order to provide better services, we should, or certain systems or other aspects, but also force the government to, uh, provide some platform for the construction of ha, this is the part, I think so. (N09) |
| Technical feedback efficiency (14) | Another is that he has some things, you say you operate wrong, you cannot change, and then it is very troublesome, uh need to report level by level. In order to delete these traces. Let's say you make one. A: No. Ah, when you make a plan, you may accidentally click the wrong point, and then it will be saved, it cannot be modified, or when you modify it will leave a trace of invalid, and then when you want to delete, you have to go up layer by layer, and not necessarily when you delete. (N23) |
| Number repetition (14) | Hm. It's being pushed across departments. Their own small program has various websites and so on. But in fact, there is a part of this function that is completely overlapping. (N35) |

*(Continued)*

**Table 3.** (Continued)

| Category (reference point/piece) | Original representative statement |
| --- | --- |
| Experience and gain (13) | Yes, it does have an impact on his way of handling complaints, that is, as a result, the complainants now do not believe in online complaints. (N24) |
| Initial issued in superior request (13) | Is the implementation of the above, yes, the beginning is the above unified requirements. (N02) |
| Time and labor costs (13) | It also relates primarily to the issue of human labor costs. (N06) |
| More publicity (12) | B: Well, the snag is that first of all, the rules of our bureau are not to be advertised. Right now, it's just peer-to-peer. (N07) |
| Policy disclosure and interpretation (11) | Yeah, that's it. It's still going to take this greater publicity, or some, uh, interpretation of the policy, uh, and then raising the awareness of the elderly, and uh. In the future, I think the need is going to be bigger and bigger. (N21) |
| Process management and processes (10) | Can only say that we now this electronic archives, it is just to scan paper materials into this to put, but not really electronic ah. (N14) |
| Platform design too idealistic (9) | We can only say that the electronic archives we have now, it is just to scan paper materials into this to put it, but not really electronic ah. (N38) |
| Regional differences (9) | Well, yes, because I used to work in the Beidaihe District office, I used to work in the smart city, Beidaihe District smart city, when I was working in that unit, I felt that the whole platform of Beidaihe District had been built very well. At that time, we managed the mayor's hotline, which is the complaint telephone of citizens. After calling the mayor's hotline, They will send a work order to our district, after sending a work order, we will be sent to the town or bureau of the territory, but I received from this side, that is, from the town of Changli this side of the paper work order, which is actually according to the efficiency is very slow, very backward. (N19) |
| Digital abuse (9) | It is too widespread. As you mentioned, this feature of the summary is quite in place, that is, everyone may be engaged in innovation and management, and if an idea comes out, it may be a platform, but it will increase the burden of administration, that is, the burden of many people. (N34) |
| Technical pressure (7) | Well, in terms of the digitization of the court, I think the system actually has more and more cases as time goes by, and then the case volume is getting higher and higher, and then the usage rate and the number of users are increasing. In fact, it is a test for the carrying capacity of the system, that is, the technology of the system. (N38) |
| Low utilization of digital products (5) | We do not have such a service such as, uh, this kind of words, we do not have ah, we are the district level, but you like the city, including the municipal party committee and the national level, it does have several platforms, there are several platforms, but the utilization rate is not high. (N09) |
| Long-term technical operation and management (5) | The implementation of this policy, he is not a quick success, that is, you can first ask for advice, and then all places can feedback what kind of interface we need, what kind of end, first build the overall framework, for example, I am, uh, just like the epidemic aspect of this instant to do the health code, the whole after the derivative will have a lot of systems, And then slowly integrate, this time will certainly not be too short, which requires a long-term problem discovery, feedback, problem solving, this spiral of this way. (N25) |
| Poor policy continuity (5) | Use, we have used before, is to use the flying book, is to use the APP, and then we deleted it, because it is no longer needed, the leader must be signed in person. (N03) |

(*Continued*)

**Table 3.** (Continued)

| Category (reference point/piece) | Original representative statement |
|---|---|
| Difference of upper and lower organization and division of labor in top group (5) | Yes or no, he is the upper level and the lower level, it is one because you, for example, his upper level he is every person, he is responsible for such a business, it must be a person, it corresponds to each business, but you go to the grassroots, then he will be responsible for a lot of people at the same time, yes, so he is a person will have a lot of UK. (N02) |
| Digital public service (5) | Ah, usually there is a problem, we go to solve the problem, coordination to solve the problem, there is no problem, then you are the enterprise's own development. (N09) |
| Management inertia (4) | I think the main problem is management, uh, the leaders don't pay enough attention, and they're used to it, that is. (N22) |
| Combine online and offline (4) | It's both. That means both online and offline are important. It's important. (N15) |
| Normalizing consciousness (3) | Well, another thing is that this is a problem of consciousness in the early stage, and later if this is normalized, in a few years, maybe everyone has this consciousness, and then it will be changed. (N12) |
| Organize training (3) | Because this is a new platform, his business is too complicated, and then it is unified organization, ah, to tell the truth, organized several times training ah, and then it is slowly so the implementation of open (N02) |
| Service object credit and compliance (3) | I know right? Well, habitually, because there are some people who are not in regular contact with taxes, yeah, you call them up, you say they're going to get a refund, they say you're a liar, it's not right, especially the smaller amount, the smaller amount, ten, eight, thirty, fifty, and then there was a problem when we were going forward, which was, uh, one was I said, let's get this straight, He then went up to operate ha, his compliance is a problem, trust is also a problem, but we internal requirements should be withdrawn, that is to say, we cannot take advantage of them this cheap, say the money that should be returned, taxpayer money ah, should be returned to him, you cannot say that the work is not good to promote ha, you do not return, we also need to assess, internal assessment. (N06) |
| Digital literacy and culture (2) | I think for example, um, we're walking, we're walking down the street and we see some municipal facility, and it might be broken or something, and it would be nice if we could, uh, have this function that reflects the problem for each of us when we see it, if we could also improve it, I think. (N28) |
| Digital cocoon room (2) | Right, so they don't have access to, you know, things that other people are interested in. It has to be more and more of what they like. (N29) |
| Digital substitution (2) | The interviewee: Well, I think there should be no such thing as a reduction in the number of people in our tax system, even though our tax system tends to be electronic, but a lot of data you just earn from the top, you earn from the system, but your data comes out, it doesn't mean that you will understand the analysis, all the things, you still have to, For example, if you call, uh, and ask the taxpayer what kind of situation, what kind of situation they have caused a current tax arrears, and some circumstances of paying taxes, then we are going to use the data, and then combined with the actual situation, for his comprehensive analysis. (N18) |

(*Continued*)

**Table 3.** (Continued)

| Category (reference point/piece) | Original representative statement |
| --- | --- |
| Digital salvage (2) | What we are doing is, uh, if your relatives and friends, they can help you with this mini application, then it is the best, because this way is more direct and more convenient, if you really can't, even if there are no relatives near your home, it is difficult to live on your own, then we will take the original way, That is, our village cadres will go to understand the situation of your family, if it is really difficult, it is directly by the village cadres to declare this matter with us, and then we go to the household to investigate. (N31) |

of policy cognition. It has a clear emotional portrait, is the derivative of cognitive environment dimension, and involves the formation of other policy implementation environments besides cognitive environment. The behavioral environment of policy implementation in China's grassroots digital construction is the sum of the activities carried out by each administrative executive body to implement the grassroots digital policy, and it involves the selection and integration process of different types of behaviors such as propaganda, coordination, system, initiative, and passivism. The normative environment and dimension of policy implementation in China's grassroots digitization construction is the sum of the agreed commonly known or expressly stipulated scales and standards that affect the implementation subjects. It is a process of guiding, standardizing and adjusting the behavior environment of policy implementation to make it gradually standardized, as well as implementing constraints and incentives to improve it. The control environment dimension of policy implementation in China's grassroots digital construction is the act and process of reviewing the existing behavioral environment dimension to control it within the normative environment dimension and gradually optimize it under the guidance of the normative environment dimension and combining with the actual situation of the stage.

Based on the above analysis, the generation model of the policy environment in China's grassroots digital construction can be constructed (Fig 1).

## 3.2 Test of theoretical saturation

The theoretical saturation test is used to indicate that when there is no new attribute in the category generated by the theoretically sampled data, the attribute of the category is saturated, and there is no new property in the theoretical category; that is, theoretical saturation is reached [28]. In this study, the remaining three policy texts were also coded and analyzed according to the rooted theory procedure to test the theoretical saturation. The results show that the categories overlap and are similar, and there are no new important categories or relationships. The categories in the model are quite rich. Therefore, it can be considered that the above model of grassroots digital policy environmental governance process is theoretically saturated.

## 3.3 Theoretical model discussion and interpretation

According to the generation model of policy implementation environment constructed via the above three-step coding, it can be seen that the cognitive dimension derives the emotional dimension, the emotional dimension influences the behavioral dimension, the behavioral dimension is the basis for the generation of the normative dimension, the normative dimension influences the behavioral dimension, and the control dimension ensures that the behavioral dimension conforms to the normative dimension. Under the mutual influence of these

**Table 4. Axial coding and category.**

| Master Category | Sub-Categories | The category that affects the relationship | The connotation of the relationship |
|---|---|---|---|
| Cognitive dimension | Self-awareness<br>External perception | Digital literacy and culture<br>Acknowledge digital Manulife and design in the first place<br>Cognition and quality of service object<br>Leadership cognition and style<br>Digital cocoon | The cognitive dimension refers to the sum of the subject's knowledge and understanding of things. It is influenced by both self-cognition and external cognition. It is the logical starting point for the generation of policy implementation environment in China's grassroots digital construction. |
| Emotional landscape dimension | Sense of Security<br>A sense of experience<br>A sense of boundaries | Data security and confidentiality<br>Digital alternatives<br>Experience and access<br>Digital public service<br>Customer credit and compliance<br>Digital rescue | The emotional dimension is the comprehensive embodiment of the subject's inner attitude towards the external things. It is composed of various feelings and feelings. It is a derivative of cognition. It constitutes the emotional link generated by the policy implementation environment in China's grassroots digital construction. |
| Behavioral environment dimension | Publicity commentary<br>Coordination and integration<br>Passive execution<br>Autonomous participation | Digital coordination and integration<br>Digital products are not practical for grassroots work<br>Grassroots autonomy<br>Digital formalism<br>The reality is an increased burden on the grassroots<br>Numbers repeat<br>Initial issued at superior request<br>Step up publicity<br>Policy disclosure and interpretation<br>Process management and procedures<br>Platform design is too idealistic<br>Regional differences<br>Number abuse<br>Low utilization of digital products<br>Top group superior and subordinate organization structure and division difference<br>Combination of online and offline<br>Organize training | The behavioral dimension is the sum of the activities produced by the subject under the influence of various internal and external stimuli. It is the active response made by the subject under certain conditions. It is the key link of policy implementation environment generation in the digital construction of China's grassroots. |
| Standard environment dimension | Legal system<br>Ideology<br>Values<br>Cultural practices | System design<br>Normalization of consciousness<br>Management inertia<br>Allocation and diversion of attention<br>Poor policy persistence | The normative dimension refers to the commonly known or expressly stipulated standards that affect the subject. It consists of the formal legal system and informal ideology, values, cultural customs and so on. It has a stimulating and constraining effect both inside and outside of the subject. It is another key link in the generation of the policy implementation environment in the digital construction of China's grassroots. |
| Control environment maintenance | Supervision and management<br>Feedback reinforcement<br>Cost control | Supervision and control<br>Technical feedback efficiency<br>Time and labor costs<br>Technical pressure<br>Long-term technical operation and management | Control environment maintenance refers to the activities and processes in which the subject is limited to a certain range by internal and external conditions. It makes the subject activity stay within a certain threshold range of activities. It is the correction and perfection of the subject's behavior. It is the standard environment and optimizes the environment to maintain it. It is an important link in the generation of policy implementation environment in China's grassroots digital construction. |

categories, the policy implementation environment is generated and gradually optimized in China's grassroots digital construction until the realization of grassroots digital goals. To be specific, their respective constituent dimensions are as follows:

**(1) The cognitive dimension.** The cognitive environment dimension is the logical starting point for the generation of policy implementation environment in China's grassroots digital construction. Under the dual influence of self-cognition and external cognition, the cognitive environment plays a leading role in the generation of policy implementation environment in grassroots digital construction. From the perspective of content composition, it consists of two categories: self-cognition and external cognition.

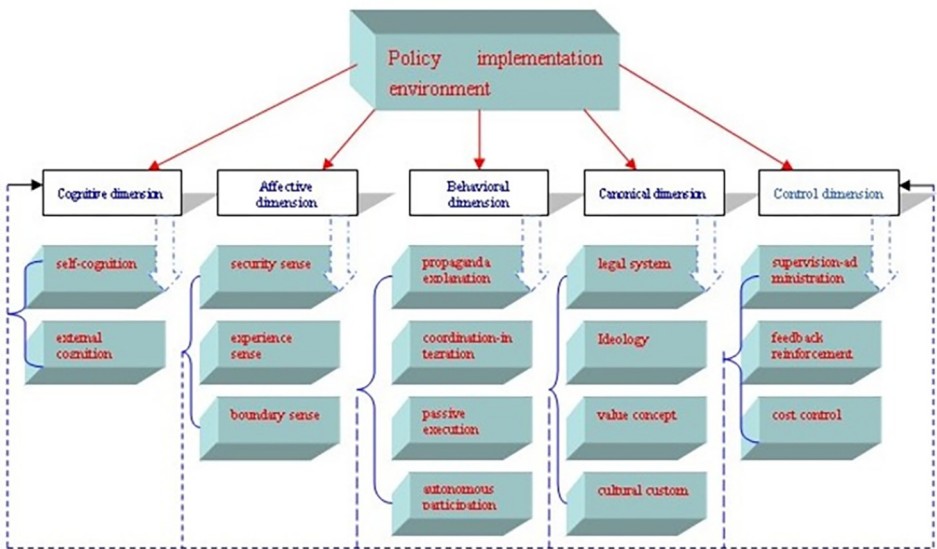

**Fig 1. Generation model of China's grassroots digital policy implementation environment.**

Self-cognition is the sum of the awareness, understanding and influencing factors of the policy implementers on their own digital environment in the face of the information revolution with digital technology as an important feature. It makes the grassroots digitization enter the vision of the policy implementation subject and is implanted into the whole process of understanding and implementing the policy. It is a necessary process of generating the policy implementation environment and an internal determining factor of sustaining the cognitive environment of policy implementation in the grassroots digitization construction. External cognition is the sum of the policy implementors' cognition and understanding of other factors affecting the implementation of grassroots digital policies besides themselves and their influencing factors. It has an important influence on the maintenance of the cognitive environment of policy implementation in the grassroots digitization construction. The external cognition plays an active and counter-effect on the internal cognition of the subject of policy implementation. When the two directions are consistent, the policy implementer will have a consistent and stable cognition and understanding of the policy environment in the grassroots digital construction, which will have a one-way strengthening effect on the policy implementation practice. When the two directions are inconsistent, the policy implementers will have contradictory knowledge and understanding of the policy environment in the grassroots digital construction, which will have a two-way impact on the policy implementation practice, and make the policy implementers face the pressure of harmonizing and balancing their own and external cognition.

In the process of interviewing the policy implementation environment in the grassroots digitization construction in China, it is found that the policy implementation subjects have a deep understanding of digital macro life and good design, believe that their digital quality and culture can gradually adapt to the needs of grassroots digitization, and believe that the quality of grassroots service objects is an important factor restricting the implementation of grassroots digitization policies. It is believed that the cognition and style of leaders have a decisive impact on the policy implementation environment in the grassroots digital construction.

**(2) The emotional environment dimension.** The emotional dimension is the comprehensive embodiment of the subject's inner attitude towards external things. It is composed of

various feelings. It is a derivative of cognition. It constitutes the emotional link generated by the policy implementation environment in China's grassroots digital construction. From the perspective of content composition, it mainly includes three aspects: one is the sense of security, the other is the sense of experience, and the third is the sense of boundary.

Security refers to the sense of confidence and freedom from fear and anxiety generated by the policy implementation subject in the face of the digital impact. It is the base and cornerstone of the emotional environment of policy implementation in China's grassroots digital construction.

The sense of experience means that the subject of policy implementation uses their own life to verify the impression left by the perception of digitalization. It is the most intuitive and key link to generate the emotional environment of policy implementation in the digital construction at the grassroots level in China.

The sense of boundary refers to the feelings of policy implementers regarding what belongs to them and what belongs to others in the digitization process at the grassroots level. It is an important link in the development of the emotional environment of policy implementation in the digital construction of the grassroots in China. It has an impact on the sense of security and experience and provides an important guarantee for the effective and orderly implementation of policies. It is also an important sign that the emotional environment of policy implementation tends to be mature in China's grassroots digital construction.

In the process of interviewing the policy implementation environment in China's grassroots digital construction, it is found that digital security and confidentiality are important concerns to maintain the emotional environment of policy implementation in grassroots digital construction. Policy implementers generally believe that the tension between the security of massive data at the grassroots level and the openness of digitalization is the key factor restricting the implementation environment of grassroots digital policy. It is believed that the sense of experience based on the sense of security is the key link to optimizing the cognitive environment and improving the behavioral environment. Only the service object has a good sense of experience; they can thus show a higher degree of credit and compliance. A clear sense of what they should do and what they should not do is an important guarantee for policy implementation subjects to accurately do or not do something and provide good digital public services.

**(3) The behavioral environment dimension.** The behavioral dimension is the sum of the activities produced by the subject under the influence of various internal and external stimuli. It is the active response made by the subject under certain conditions. It is a key link in the formation of the policy implementation environment in China's grassroots digital construction, because it is the activity and process in which the policy implementation subject takes action to implement the policy under the stimulation and influence of the cognitive environment and the emotional environment. It is an important reference link to regulate and control the environment of peacekeeping. Its actual process is a test of whether the policy implementation environment is conducive to policy implementation, and it will also have an impact on the emotional environment of cognitive and peacekeeping. It plays an important role in the whole process of generating the policy implementation environment in China's grassroots digital construction. It is the most active, uncertain and complex key link in the generation of the whole policy implementation environment.

In the process of interviewing the policy implementation environment in China's grassroots digital construction, it is found that the grassroots digital behavior initially emerges from the arrangements and requirements of the upper level, and the grassroots autonomy is not strong. Currently, it is mostly carried out by the combination of online and offline, but it is subject to the traditional process and there is still an idealized platform design. Problems include the mismatch between grassroots products and grassroots work, digital formalism and digital abuse,

the low overall utilization rate of digital products, being limited by regional differences, widespread imbalance in grassroots digital development, and an urgent need to strengthen holistic and coordinated governance. In addition, grassroots digital policies are not popular and well-informed at the grassroots level. Many grassroots policy implementors are not clear about China's macro, medium and micro digital policies; therefore, we should improve the publicity and interpretation of policies, create a good behavior environment, and help improve the efficiency of grassroots digital policy implementation.

**(4) Standardizing the environment dimension.** The normative dimension refers to the standards that affect the subject. It consists of the formal legal system and informal ideology, values, cultural customs, and so on. It has a stimulating and constraining effect both inside and outside the subject. It is another key link in the generation of the policy implementation environment in the digital construction of China's grassroots. It not only provides important standards and guidance for the generation of behavioral environment but also the soil infiltrated by the cognitive environment, emotional environment and behavioral environment. In terms of the implementation of policies in China's grassroots digital construction, its normative environment is different from and related to the whole grassroots digital construction policy environment. On the one hand, the normative environment of policy implementation is part of the hard standard system and soft historical and cultural factors related to the actual implementation in the whole policy environment, and it is the focus and internalization of the whole policy environment in the implementation level; on the other hand, the standard environment of policy implementation is restricted and affected by the overall policy environment, and the two interact with each other.

In the process of interviewing the policy implementation environment in China's grassroots digital construction, it is found that unreasonable system design, poor policy continuity caused by shifts in government attention allocation, and the inertia of existing grassroots policy implementation are the main factors for the optimization and improvement of the regulatory environment in grassroots digital construction.

**(5) Controlling the environment dimension.** The control dimension refers to the activities and processes in which the subject is restricted to a certain range by internal and external conditions. It keeps the subject activity within a certain threshold. It is the correction and perfection of the subject's behavior. It is the continuation link of the standard environment and the behavior environment. It is also an important link in the generation of the policy implementation environment in China's grassroots digital construction. It is the inflection point of closed-loop management of the policy implementation environment in China's grassroots digital construction within a certain policy time limit. Therefore, it is not only the end point but also the starting point, or the link that is associated with the cognitive environment dimension, emotional environment dimension, behavioral environment dimension, normative environment dimension and every other policy environment link at any time. It is also a very important link. This link governance is not good; it is easy for policy implementation behavior deviation to emerge, seriously affecting the optimization and improvement of the policy implementation environment, as well as leading to grassroots digital suspension. Grassroots digital suspension refers to a situation in which digital integration at the grassroots level does not truly occur; instead, it is separated from grassroots governance outside or parallel to the traditional grassroots governance state. It will bring digital formalism, cause digital burdens, and cannot truly realize digital empowerment at the grassroots level. This should be given attention in the implementation of China's grassroots digital policy in the future to avoid this phenomenon as much as possible.

In the interview process of the policy implementation environment in China's grassroots digital construction, it is found that the generation of a long-term benign control environment

requires the investment of a lot of human resources, material resources, financial resources and intelligence, which is a very challenging task at present and in the future, especially in the current post-pandemic era. How do we coordinate and balance the relationship between economic development and long-term digital construction? This an important question to answer.

## 4. Clear set qualitative comparative analysis

As one of the qualitative comparative analysis methods, clear set qualitative comparative analysis (csQCA) is applicable to 10–60 samples. Based on the further verification and extension of the research results of rooted theory, this study takes 37 interview texts with rooted coding as the research object to further investigate the executive power of the policy implementation environment in China's grassroots digital construction. The implementation of the policy implementation environment referred to in this study refers to the ability of the policy implementation subject to evaluate whether the policy implementation environment in which it is located is conducive to policy implementation. That is to say, if the positive evaluation of the policy implementation environment is greater than the negative evaluation, it is considered that the policy implementation environment in which the policy implementation subject is located has a strong policy implementation ability. This part is mainly realized through the emotional automatic coding function of NVivo12plus.

### 4.1 Setting of explanatory variables

Based on the previous rooted coding analysis results, this study takes five generative elements as explanatory variables of the policy environment generation model in China's grassroots digital construction. First, with the help of the matrix coding function of NVivo12plus, a coding matrix with 37 interview text cases as rows and 5 generative elements as columns is obtained (see Table 5). Then, under the "bipartite attribution principle" method, the mean value method is used to assign "1" and "0" to the explanatory variables.

### 4.2 Setting of the result variable

Based on the results of the rooted coding analysis above, this study considers whether the policy implementation environment, as evaluated via the policy implementation subject, is conducive to policy implementation as the criterion; that is, when the policy implementation subject's positive evaluation of its policy environment is greater than the negative evaluation, it is assigned 1, whereas when a negative evaluation predominates, it is assigned as 0. This part uses the automatic coding function of NVivo12plus for emotion recognition and obtains the calculation value of reference point according to very positive * (1) + relatively positive * (0.7) + relatively negative (-0.7) + very negative * (1). If the calculation value is greater than 0, the value is assigned as 1; otherwise, it is assigned as 0. The interview texts numbered NB, NC, NH, NI, NG, NM, NO, NQ, NR, NU, NX, NY, NZ, Nb, Nc, Nf, Ng, Nh, and Ni are 1 (the corresponding coding reference points are shown in Table 6), and the other values are 0.

### 4.3 Truth table construction

After assigning values to explanatory and outcome variables, the truth table is built according to QCA's research steps. According to the assignment principles of explanatory variables and outcome variables in each of the above cases, we summarized the data and obtained the truth table, as shown in Table 7. The truth table code of explanatory variable in case NB is 01100 (from left to right in Table 7), which indicates that the environmental effect of policy

**Table 5. Setting of explanatory variables for digitization policy at the grassroots level.**

| Id. | Cognitive context dimension | Emotional context | Behavioral context | Normative context | Control dimension |
|-----|-----|-----|-----|-----|-----|
| NA | 2 | 0 | 0 | 0 | 1 |
| NB | 2 | 2 | 8 | 0 | 1 |
| NC | 15 | 0 | 8 | 6 | 0 |
| ND | 11 | 3 | 14 | 0 | 0 |
| NE | 0 | 5 | 18 | 8 | 3 |
| NF | 5 | 4 | 6 | 2 | 5 |
| NG | 4 | 3 | 3 | 1 | 0 |
| NH | 21 | 6 | 7 | 0 | 1 |
| NI | 5 | 1 | 12 | 1 | 0 |
| NJ | 8 | 5 | 3 | 0 | 0 |
| NK | 4 | 0 | 15 | 1 | 4 |
| NL | 0 | 0 | 12 | 6 | 8 |
| NM | 8 | 0 | 15 | 0 | 5 |
| NN | 6 | 2 | 4 | 0 | 4 |
| NO | 1 | 3 | 5 | 0 | 3 |
| NP | 8 | 0 | 3 | 5 | 7 |
| NQ | 5 | 2 | 6 | 0 | 1 |
| NR | 4 | 1 | 5 | 1 | 0 |
| NS | 3 | 0 | 3 | 1 | 0 |
| NT | 2 | 0 | 4 | 0 | 4 |
| NU | 6 | 1 | 4 | 1 | 1 |
| NV | 3 | 2 | 13 | 3 | 2 |
| NW | 4 | 3 | 14 | 0 | 2 |
| NX | 7 | 0 | 13 | 2 | 4 |
| NY | 5 | 2 | 2 | 0 | 3 |
| NZ | 6 | 2 | 2 | 4 | 3 |
| Na | 9 | 0 | 4 | 3 | 1 |
| Nb | 7 | 2 | 7 | 1 | 0 |
| Nc | 7 | 0 | 1 | 0 | 0 |
| Nd | 8 | 2 | 1 | 2 | 0 |
| Ne | 3 | 0 | 8 | 1 | 1 |
| Nf | 7 | 0 | 4 | 0 | 2 |
| Ng | 1 | 1 | 2 | 0 | 1 |
| Nh | 3 | 0 | 5 | 2 | 1 |
| Ni | 4 | 0 | 1 | 0 | 3 |
| Ng | 3 | 0 | 6 | 1 | 1 |
| Nk | 5 | 0 | 4 | 0 | 3 |
| Average | 5.46 | 1.41 | 6.54 | 1.41 | 2.03 |

implementation of grassroots digital construction in case NB is mainly generated by the emotional environment dimension and behavioral environment dimension. The truth table code of the result variable of case NB is 1, which means that the environmental configuration produces a good environmental effect, so the result variable is 1.

## 4.4 Research results and analysis

**4.4.1 Univariate necessity analysis.** Qualitative comparative analysis is used to study cases of a "multiple causes and one effect" nature, so it is necessary to test whether each factor

**Table 6. Basis for setting the result variables.**

| Id. | Interview subjects | Reference Point Accounting Values |
|---|---|---|
| NB | 02 Kings * | 1.4 |
| NC | 03 Liu * | 4.8 |
| NH | 08 Lee **+13 Gow ** | 2.2 |
| NI | 09 Liu ** | 4.3 |
| NJ | 10 sheets ** | 0.6 |
| NM | 16 Liu * | 1.3 |
| NO | 18 Hu ** | 3.9 |
| NQ | 20 levels ** | 6 |
| NR | 21 holes ** | 6.3 |
| NU | 24 Wu ** | 0.8 |
| NX | 27 sheets ** | 9.9 |
| NY | 28 Xu ** | 8.1 |
| NZ | 29 beams * | 1.1 |
| Nb | 31 Pay ** | 2.4 |
| Nc | 32 single ** | 3.2 |
| Nf | 35 Liu * | 0.3 |
| Ng | 36 Liu * | 1.1 |
| Nh | 37 Lee ** | 2.2 |
| Ni | 38 in ** | 2.8 |

can independently affect the outcome. If each explanatory variable in the case can independently affect the outcome variable, the QCA method cannot be used in this case [29]. Therefore, it is necessary to conduct a univariate necessity analysis on the condition variables of the selected cases.

After the truth table is constructed, the "XY plot" is run in Tosmana software to test whether there is a single explanatory variable influencing the result variable in the case, that is, to analyze whether there is a sufficient or necessary relationship between a single explanatory variable and the result variable. Consistency and coverage are used to test the above relationship in the QCA method.

The judgment principle of consistency is as follows: if the consistency is greater than 0.8 and less than 0.9, it indicates that the single conditional variable X is a sufficient condition for the result variable Y; if the consistency is greater than 0.9, it means that the single variable X is a necessary condition for the result variable Y, and vice versa. The coverage rate is used to explain which combination of condition variables has explanatory power for outcome variable Y. The greater the coverage rate, the more reason X has to explain Y.

As shown in Table 8, the consistency of the five condition variables is below 0.8, which indicates that these five variables cannot independently affect the grassroots digital policy implementation. The grassroots digital policy implementation is the result of the joint action of multiple policy factors, rather than being determined by a single policy factor. It is necessary to find an effective combination of variables to optimize the environmental effect of policy implementation in China's grassroots digital construction.

**4.4.2 Qualitative comparison results and theoretical analysis.** After the necessity test of a single variable is passed, the truth table in EXCEL is saved in the.csv file format, the saved.csv file is imported to Tosmana, and "Start csQCA" is run to obtain the combination of policy elements that affect the policy implementation environment in the grassroots digital construction. After calculation, Tosmana software obtains four configuration presentation modes,

**Table 7. Truth table of variable combinations in different interview cases.**

| Case | Cognitive Context Dimension (CO) | Emotional Jingwei (EM) | Behavioral Dimension (BE) | Specification Boundary Dimension (ST) | Control Border Dimension (CT) | Result Environmental Effects (EF) |
|---|---|---|---|---|---|---|
| NA | 0 | 0 | 0 | 0 | 0 | 0 |
| NB | 0 | 1 | 1 | 0 | 0 | 1 |
| NC | 1 | 0 | 1 | 1 | 0 | 1 |
| ND | 1 | 1 | 1 | 0 | 0 | 0 |
| NE | 0 | 1 | 1 | 1 | 1 | 0 |
| NF | 0 | 1 | 0 | 1 | 1 | 0 |
| NG | 0 | 1 | 0 | 0 | 0 | 0 |
| NH | 1 | 1 | 1 | 0 | 0 | 1 |
| NI | 0 | 0 | 1 | 0 | 0 | 1 |
| NJ | 1 | 1 | 0 | 0 | 0 | 1 |
| NK | 0 | 0 | 1 | 0 | 1 | 0 |
| NL | 0 | 0 | 1 | 1 | 1 | 0 |
| NM | 1 | 0 | 1 | 0 | 1 | 1 |
| NN | 1 | 1 | 0 | 0 | 1 | 0 |
| NO | 0 | 1 | 0 | 0 | 1 | 1 |
| NP | 1 | 0 | 0 | 1 | 1 | 0 |
| NQ | 0 | 1 | 0 | 0 | 0 | 1 |
| NR | 0 | 0 | 0 | 0 | 0 | 1 |
| NS | 0 | 0 | 0 | 0 | 0 | 0 |
| NT | 0 | 0 | 0 | 0 | 1 | 0 |
| NU | 1 | 0 | 0 | 0 | 0 | 1 |
| NV | 0 | 1 | 1 | 1 | 0 | 0 |
| NW | 0 | 1 | 1 | 0 | 0 | 0 |
| NX | 1 | 0 | 1 | 1 | 1 | 1 |
| NY | 0 | 1 | 0 | 0 | 1 | 1 |
| NZ | 1 | 1 | 0 | 1 | 1 | 1 |
| Na | 1 | 0 | 0 | 1 | 0 | 0 |
| Nb | 1 | 1 | 1 | 0 | 0 | 1 |
| Nc | 1 | 0 | 0 | 0 | 0 | 1 |
| Nd | 1 | 1 | 0 | 1 | 0 | 0 |
| Ne | 0 | 0 | 1 | 0 | 0 | 0 |
| Nf | 1 | 0 | 0 | 0 | 0 | 1 |
| Ng | 0 | 0 | 0 | 0 | 0 | 1 |
| Nh | 0 | 0 | 0 | 1 | 0 | 1 |
| Ni | 0 | 0 | 0 | 0 | 1 | 1 |
| Ng | 0 | 0 | 0 | 0 | 0 | 0 |
| Nk | 0 | 0 | 0 | 0 | 1 | 0 |

Source: The author based on the actual situation of 37 cases.

namely, configuration 1, configuration 0, configuration C and configuration R (Fig 2). The so-called configuration 1 is the sum of the combination of policy elements that make the policy implementation environment produce a greater effect; configuration 0 refers to the sum of the combination of policy elements that do not produce a greater effect on the policy implementation environment; configuration C is the contradictory configuration; configuration R is the logical remainder. The combined path of configuration 1 and configuration 0 is shown in Table 8.

**Table 8. Univariate consistency and coverage.**

| Name of Variable | consistency (consistency) | coverage |
|---|---|---|
| Cognitive Dimension (CO) | 0.6667 | 0.5263 |
| Emotional Context Dimension (EM) | 0.5000 | 0.4211 |
| Behavioral Context Dimension (BE) | 0.5000 | 0.3684 |
| Specification Dimension (ST) | 0.3636 | 0.2105 |
| Control Dimension (CT) | 0.4286 | 0.3158 |

Data source: compiled according to the calculation results of the software Tosmana.

In particular, the result of configuration 1 is shown as follows:

Result = CO * em * BE * ST+CO * be * st * ct+CO * em * BE * CT+co * EM * be * st * CT +CO * EM * be * ST * CT+co * em * be * ST * ct

The results of configuration 0 are expressed as follows:

Result = co * EM * ST * CT +co * EM * BE * ST +co * em * BE * CT+CO * em * be * ST +CO * be * ST * ct+CO * EM * be * st * CT

(the "*" in the formula represents the "and" in the intersection, with uppercase for presence and lowercase for absence)

As can be seen from the qualitative comparison results in Table 9, Path 2 has the highest coverage rate, indicating that compared with other paths, Path 2 is a combination mode of factors that play a role in the environmental effect of policy implementation in China's grassroots digital construction. The corresponding interview cases are NJ, NU, Nc and Nf, that is, 10 interviewees, 24 Wu **, 32 single ** and 35 Liu *. Their main views are shown in Table 10, indicating that, currently, the cognitive dimension plays a leading role in the policy

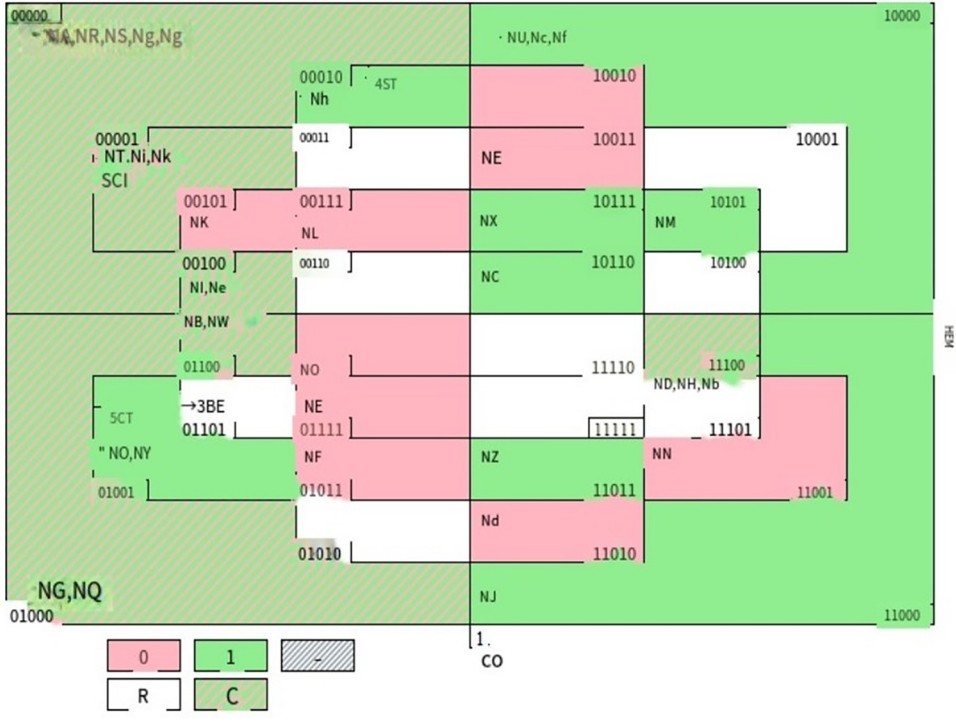

**Fig 2. Results of qualitative comparative analysis.**

**Table 9. Analysis of qualitative comparison results of combination paths of different policy environment effect factors.**

| Path combination | Coverage | Consistency (consistency) |
|---|---|---|
| Path 1 CO * em * BE * ST | 0.1053 | 1.0000 |
| Path 2 CO * be * st * ct | 0.2105 | 1.0000 |
| Path 3 CO * em * BE * CT | 0.1053 | 1.0000 |
| Path 4 co * EM * be * st * CT | 0.1053 | 1.0000 |
| Path 5 CO * EM * be * ST * CT | 0.0526 | 1.0000 |
| Path 6 co * em * be * ST * ct | 0.0526 | 1.0000 |

implementation environment of China's grassroots digital construction, while the environmental effects of behavioral dimension, normative dimension and control dimension have not been well utilized; that is, overall, China's grassroots digital construction is still in its initial stage. Most of the grassroots policy implementation subjects can have a fuller understanding of the digital empowerment of grassroots development, but subject to their low digital quality, grassroots digitization has not been developed in a real sense, and the cognition and style of grassroots leaders have a great impact on the grassroots digital policy implementation environment. At present, there are problems of digital fragmentation, digital duplication and digital formalism at the grassroots level.

On the whole, the six paths obtained by the clear set qualitative comparative analysis in this study are the main combination paths for the policy environmental effect to play in the grassroots digital construction in China. Other specific analyses are as follows:

Path 1: Knowing-doing-normative type—its result formula is (cognitive dimension) * (~ emotional dimension) * (behavioral dimension) * (normative dimension) (that is, CO * em * BE * ST, where "~" represents not, equal to the combination of lowercase letters).

Path 1 represents that the cognitive dimension, behavioral dimension and norm dimension of policy execution can be generated in China's grassroots digital construction, so that the

**Table 10. Cases and main points of view corresponding to Path 2.**

| Interview Cases | Interviewee | Main point |
|---|---|---|
| NJ | 10 sheets ** | Digital governance breaks through the limitations of time and space and improves the work efficiency to a certain extent. Leaders are older and prefer paper versions of reporting materials, which is a release of power for leaders. |
| NU | 24 Wu ** | The degree of digitization of the district and county petition work is relatively high, the standardization of the online petition standardization process is lower than that of the offline process, and the grassroots work subjects below the district and county level are not strong enough in their awareness and autonomy to improve their own digital level and quality. They should increase the publicity, promotion and utilization rate of digital products, and arouse the awareness of grassroots staff to participate in digital governance with a sense of experience and acquisition. |
| Nc | 32 single ** | The high degree of digitization of the urban management system acknowledges the convenience brought by digitization to the work, and the management and service interfaces have tried multiple digital products. Welcome the further digital transformation of the urban management work and recognize that the lack of professional talents is the biggest constraint encountered in the future digital governance efficiency optimization. |
| Nf | 35 Liu * | The platform for connecting trust, law and discipline has been running well for more than a year, with problems such as insufficient publicity and digital formalism. The town government has its own website, which can handle business. There are problems of digital duplication and digital fragmentation, which affect the efficiency of grassroots governance to a certain extent and cause the waste of grassroots human resources, material resources and financial resources. |

**Table 11. Cases and main views corresponding to Path 1.**

| Interview Cases | Interviewee | Main point |
|---|---|---|
| NC | 03 Liu * | The shift in attention results in the waste of digital resources at the grassroots level, and the reuse rate of digital resources is not high. Leaders tend to follow the existing management practices and have no motivation for digital application and innovation. The digital quality of leaders restricts the digitization at the grassroots level, the penetration and impact of digital power and traditional power, and the transformation and impact of digital scenes and traditional scenes all affect the digitization process at the grassroots level. |
| NX | 27 sheets ** | The attention of the Tourism and Culture Radio and Television Bureau is mainly allocated to the business side rather than the government side, the tourism business interface has a high degree of digitalization, and digitally enabled tourism has achieved certain results. However, the tension between digital governance and traditional governance still exists, the flatness required for digital governance has not been realized, the phenomenon of mismatch between digital products and the actual situation at the grassroots level still exists, the module design of digital products urgently needs to be optimized, and the integration and coordination of digital needs to be further promoted. |

positive environmental effect of policy can be better realized. In other words, the combination of the cognitive environment dimension, behavioral environment dimension and normative environment dimension is an effective path to exert the environmental effect of policy implementation in China's grassroots digital construction. The corresponding interview cases of this path are NC and NX, namely interviewees "03 Liu *" and "27 Zhang **". Their main views are shown in Table 11. This path shows the following: On the one hand, leaders' digital cognition and quality is the main cognitive dimension obstacle restricting grassroots digital construction at present. The root cause is that the relationship and transformation between digital power and traditional organizational power and digital scene and traditional scene are not straightening out. How to straighten out the relationship between the two is an important issue to address in the future. On the other hand, on the whole, the digitization process of the grassroots business side is faster than that of the government side. Although under the guidance of the top-level design of the state, the grassroots have carried out a considerable degree of behavioral exploration, and there are still some problems such as digital products not being grounded, digital fragmentation being serious, and the organizational structure not being flat enough. In addition, in view of the security and confidentiality of traditional government activities, the policy implementer's sense of security, experience and boundary of digital-related change activities has not been truly generated, or the current policy implementer is optimistic and prudent about grassroots digitalization, and more efforts should be made to create a good emotional environment in the future.

Path 2: cognitive-oriented—its result formula is (cognitive dimension) * (~ behavioral dimension) * (~ normative dimension) * (~ control dimension) (CO * be * st * ct).

Path 3: Know-doing-control type—its result formula is (cognitive dimension) * (~ emotional dimension) * (behavioral dimension) * (control dimension) (CO * em * BE * CT).

Path 3 indicates that even without the generation of the emotional dimension, under the combination of the cognitive dimension, behavioral dimension and control dimension, the policy implementation environment can still play an effect in the digital construction of the grassroots level in China. The interview cases corresponding to this path are NM and NX, that is, interviewees "16 Liu *" and "27 Zhang **". Their main views are shown in Table 12. This path shows the following:: On the one hand, at present, grassroots digitization has achieved certain results under the functions of the cognitive, behavioral and control dimensions, but the

**Table 12. Cases and main views corresponding to Path 3.**

| Interview Cases | Interviewee | Main point |
|---|---|---|
| NM | 16 Liu * | The level of digital governance needs to be improved, and the issue of digital fragmentation urgently needs to be addressed. |
| NX | 27 photos ** | The attention of the Tourism and Culture Radio and Television Bureau is mainly focused on the business side rather than the government side, the tourism business interface has a high degree of digitalization, and digitally enabled tourism has achieved certain results. However, the tension between digital governance and traditional governance still exists, the flatness required for digital governance has not been realized, the phenomenon of mismatch between digital products and the actual situation at the grassroots level still exists, the module design of digital products urgently needs to be optimized, and the integration and coordination of digitalization needs to be further promoted. |

level of grassroots digitization governance is not high enough, and the matching degree between digitalization and grassroots is not enough. How to promote the optimization of control dimension and improve the adaptability of digital products to the grassroots is an important focus of policy implementation environment optimization in China's future grassroots digitization construction. On the other hand, at the grassroots level where the emotional environment dimension of the main body of the policy implementation has not been generated, strengthening the top-level design, using the holistic governance theory and synergy theory, optimizing the behavioral environment dimension, promoting the flat governance structure, and breaking the tension between digital governance and traditional governance constitute an important path for the environmental optimization of the grassroots digital policy implementation.

Path 4: Emotion-control type—its result formula is (~ cognitive dimension) * (emotional dimension) * (~ behavioral dimension) * (~ normative dimension) * (control dimension) (co * EM * be * st * CT).

Path 4 indicates that in the digital construction at the grassroots level in China, in the absence of a cognitive environment, behavioral environment and normative environment of peacekeeping, only a good emotional environment and control environment are created, and the policy implementer can also receive help from the policy environment and enjoy the empowerment of the policy implementation environment. The interview cases corresponding to this path are NO and NY, that is, interviewees "18 Hu **" and "28 Xu **". Their main views are shown in Table 13. This path shows that digitalization is an instrumental trend of change for the grassroots policy implementation subject; even if they do not understand the global

**Table 13. Cases and main views corresponding to Path 4.**

| Interview Cases | Interviewee | Main point |
|---|---|---|
| NO | 18 Hu ** | The overall degree of digitalization of tax administration is relatively high, acknowledging the benefits and convenience brought by digitalization, but there are currently problems such as digital products being too stereotyped and not flexible enough and digital platforms not providing timely feedback and governance. I am not worried about the employment pressure brought by digitalization because I feel that digital products cannot really understand complex and uncertain management factors and do not have a substitute role for labor. |
| NY | 28 Xu ** | The document circulation system is more and more convenient than before, the digital urban management has a high degree of recognition that digital quality and literacy need to be improved, digital products should be promoted, and the awareness and utilization rate should be increased. |

digitalization wave, digital China, network power and other national strategies, the technical attributes and tool principles of digitalization itself are not clear; that is, they do not have a good understanding of grassroots digitalization. As long as they satisfy their emotional demands for the sense of security, experience and boundary of technological tools, and provide them with technical feedback guidance and cost control, grassroots policy implementers will spontaneously use digital technology to carry out work in order to complete their work more conveniently, that is, they will be manifested as personal spontaneous digitization in terms of emotional demands. In other words, at present, many grassroots organizations and individuals are driven by the good emotional benefits of digitalization and spontaneously carry out digital exploration of unit affairs or business work. For example, administrative examination and approval departments voluntarily purchased digital products to improve the quality of government services many years ago, and they enjoyed certain digital benefits in this process. However, spontaneous digital governance has also brought about the predicament of a low digital utilization rate at the grassroots level. Therefore, improvements should be made in the optimization of behavioral and normative dimensions such as strengthening the publicity and promotion of digital products, improving the awareness of digital policies as well as the utilization rate of digital products.

Path 5: Behavior deficiency type—its result formula is (cognitive dimension) * (emotional dimension) * (~ behavioral dimension) * (normative dimension) * (control dimension) (CO * EM * be * ST * CT).

Path 5 indicates that even if the behavioral dimension is not really generated, the combination of the cognitive dimension, emotional dimension, normative dimension and control dimension can still drive the implementation environmental effect of policies in the grassroots digital construction of China to a certain extent. In other words, the current phenomenon of wait-and-see change or formal implementation exists in some grassroots digital construction in China. Even under the combination of the cognitive environment dimension, emotional environment dimension, normative environment and control environment dimension of peacekeeping, the grassroots still has the problem of a lack of behavioral environment dimension of policy implementation, which restricts the level and quality of grassroots digital construction to a large extent. The interview case corresponding to this path is NZ, the interviewer "29 Liang *". She said in the interview that "the grassroots lack autonomy in the continuous updating of platform content, the traditional consciousness and cognition of leaders affect the digital innovation of grassroots work, the digital cocoon restricts people's digital innovation, and the level of grassroots digital governance is very limited and needs to be improved." This path shows that the lack of behavioral dimension is still an important constraint on the effect of policy implementation environment in the process of grassroots digital construction in China, and grassroots digitalization is faced with the dual behavioral dimension dilemma of passive formal implementation and insufficient active participation management.

Path 6: norm-dominated—its result formula is (~ cognitive dimension) * (~ emotional dimension) * (~ behavioral dimension) * (normative dimension) * (~ control dimension) (co * em * be * ST * ct).

Path 6 indicates that the normative dimension plays an important leading role in China's grassroots digital construction. Even though the grassroots policy implementer does not fully recognize and understand the attributes of digital technology, it does not utilize digital construction activities in a real sense in order to generate a better sense of security, experience and boundary in the process of digitization. In the absence of a good technical feedback and supervision and assessment environment at the grassroots level, there will still be a certain degree of environmental effect of policy implementation under the guidance of top-level system design and planning. The interview case corresponding to this path is Nh, the interviewer "37 Li **".

**Table 14. Analysis of qualitative comparison results of combination paths of different policy environment effects (0 configuration).**

| Path combination | Coverage | Consistency (consistency) |
|---|---|---|
| ~ Path 1 co * EM * ST * CT | 0.1111 | 1.0000 |
| ~ Path 2 co * EM * BE * ST | 0.1111 | 1.0000 |
| ~ Path 3 co * em * BE * CT | 0.1111 | 1.0000 |
| ~ Path 4 CO * em * be * ST | 0.1111 | 1.0000 |
| ~ Path 5 CO * be * ST * ct | 0.1111 | 1.0000 |
| ~ Path 6 CO * EM * be * st * CT | 0.0556 | 1.0000 |

She said in the interview: "Under the arrangement and guidance of the superior, the town government has carried out certain digital exploration, which has saved the time and labor cost of grassroots governance. However, the existing digital system lacks integration and coordination, and the grassroots digitization does not match the actual grassroots governance. The quality of digital service at the grassroots level cannot meet the actual situation at the grassroots level, but she is still optimistic about the further implementation of digitalization." The path shows that China's grassroots digital policy implementation environment has the characteristics of strong obedience, strengthening top-level design, and paying attention to integration and coordination, and it is an important path for China's grassroots digital construction and development.

As can be seen from Fig 2, green represents configuration 1 and pink represents configuration 0. Different from configuration 1, the path combinations that can exert the environmental effects of policy implementation are obtained, while configuration 0 shows which combination paths of policy environment elements cannot exert the environmental effects of policy implementation. On the whole, this study uses clear set qualitative comparative analysis to find six policies that hinder the implementation of policy environment to exert its effects (see Table 14). The specific analysis is as follows:

~ Path 1: (~ cognitive dimension)* (affective dimension)* (normative dimension)* (control dimension) (i.e., co * EM * ST * CT)

~ Path 1 indicates that in the current digital construction at the grassroots level in China, even if the emotional environment dimension, normative environment and control environment are generated, the cognitive environment dimension of the grassroots policy implementation subject is not generated, and the policy implementation environment cannot play a positive role in the policy implementation practice. This path indicates that the cognitive environment dimension is very important in the initial stage of grassroots digital construction in China, and the lack of a good cognitive environment dimension will limit the environmental effect of policy implementation. If the grassroots policy implementer sees digitization as a task, or even a formalism to use for the sake of use, digital governance will become a pressure governance rather than an enabling governance, and the grassroots digitization will lose its meaning. Therefore, it is crucial to grasp the cognitive bias barrier and create a good cognitive environment for the grassroots policy implementer. The interview cases corresponding to this path are NE and NF, that is, the interview groups "05 Meng **+11 Song **" and "06 Wang *+12 Lin **". Their main views are shown in Table 15. From their views, it is not difficult to see that their cognition of digitization at the grassroots level is neither dialectical nor adequate, and even contradictory, and a positive cognitive landscape has not been generated.

~ Path 2: (~ cognitive dimension)* (emotional dimension)* (behavioral dimension)* (normative dimension) (co * EM * BE * ST)

~ Path 2 indicates that if a grassroots organization does not generate the cognitive dimension of policy implementation in digital construction, then even if a certain emotional

**Table 15. The cases and main views corresponding to Path 1.**

| Interview Cases | Interviewee | Main point |
|---|---|---|
| NE | 05 Meng **+11 Song ** | Digitalization is more about tasks, use for use's sake, inadequate applicability of digital products, the existence of digital formalism, and the notion that digital governance has become digital pressure, and the contradiction between the openness of digital governance, the complexity of grassroots work, and the improvement of grassroots service quality is prominent. The digital scene is not yet suitable for the grassroots scene, and the grassroots tend to execute and have no motivation to take the initiative. |
| NF | 06 King *+12 Lin ** | The residue of digital governance is conducive to both supervision and protection of cadres. Digitalization is imperative, there are obstacles in technology and organizational structure, superiors have supervision and inspection of digital platforms, the tax system has realized digital personnel, and the future experience governance will reflect governance effectiveness. |

dimension, behavioral dimension and normative dimension is generated, the positive role of policy implementation environment cannot be played. In other words, for grassroots organizations in the initial stage of digitization, the construction of cognitive environment is a basis point, and other environments can better exert their positive effects only based on the cognitive environment dimension, because the subject of policy implementation is people, and only they can truly know and understand why grassroots digitization is necessary, what grassroots digitization can bring to grassroots, and how grassroots digitization can be achieved. In the process of digital transformation, they can overcome various problems in the transformation period and ensure that the digital policy of the grassroots can really be implemented. The interview cases corresponding to this path are NE and NV, namely the interview group "05 Meng **+11 Song **" and the interviewee "25 Zhao **". Their main views are shown in Table 16. From their views, it can be seen that they hold a pessimistic attitude towards digitization at the grassroots level, believing that digitization at the grassroots level is more formalistic and not suitable for the actual situation at the grassroots level at all. In particular, they note that it is difficult for the township and below to have the conditions to realize digitization. If the policy implementers carry out digital reform in such a cognitive environment, it can be expected that the governance effect will certainly be poor.

~ Path 3: (~ cognitive dimension)* (~ emotional dimension)* (behavioral dimension)* (control dimension) (i.e. co * em * BE * CT)

**Table 16. Cases and main views corresponding to Path 2.**

| Interview Cases | Interviewee | Main point |
|---|---|---|
| NE | 05 Meng **+11 Song ** | Digitalization is more about tasks, use for use's sake, inadequate applicability of digital products, the existence of digital formalism, and the notion that digital governance has become digital pressure, and the contradiction between the openness of digital governance, the complexity of grassroots work, and the improvement of grassroots service quality is prominent. The digital scene is not yet suitable for the grassroots scene, and the grassroots tend to execute and have no motivation to take the initiative. |
| NV | 25 Zhao ** | The grassroots digital fragmentation phenomenon is serious, digital coordination and integration is more difficult, grassroots digitalization urgently needs to strengthen the top-level design and high-level overall coordination and management, the township and below digital governance is difficult, there is no digital governance of human and financial conditions, and digital security and security, digital governance professionals, and the debugging of existing personnel are all important issues of grassroots digitalization. |

~ Path 3 indicates that the grassroots level only provides a good behavioral environment and control environment; without generating a good cognitive environment and emotional environment, it is impossible to create a policy implementation environment that has a positive stimulating effect on the policy implementation subjects. In other words, although the grassroots has carried out digital reform, has formed a certain behavioral environment of digital policy implementation, and has carried out certain control activities such as internal assessment on it, the policy implementers have poor knowledge and understanding of the digitization of the grassroots, and have not developed a positive attitude toward the existing digital governance. The soft environment for policy implementation of grassroots digital construction is thus still not conducive to the implementation of relevant policies. The interview cases corresponding to this path are NK and NL, that is, the interviewees "14 Zhang **" and "15 Zang **". Their main views are shown in Table 17. From their views, it can be seen that the grassroots units of the two interviewees have carried out a certain degree of digital exploration, resulting in a certain behavioral environment. Moreover, supervision and assessment were conducted with the help of digital means to generate a certain control dimension, but their cognitive and emotional evaluation of their past practice is not high. If other policy implementation subjects also share the same cognitive and emotional dimension, it is difficult to improve the overall digital efficiency of the grassroots unit. This path shows that, on the one hand, the cognitive environment and emotional environment, as the soft environment of policy implementation environment in grassroots digital construction, will change under the influence of the behavioral environment and control environment, and a bad cognitive environment and emotional environment will hinder the long-term digitization process of grassroots units. For the grassroots organizations that have created the digital policy and behavior environment and peacekeeping control environment, it is especially necessary to strengthen the construction of a standardized environment by improving the system and regulations, as well as to pay attention to the cultivation and guidance of the cognitive environment and emotional environment of the policy implementation subjects.

~ Path 4: (cognitive dimension)* (~ emotional dimension)* (~ behavioral dimension)* (normative dimension) (i.e. CO * em * be * ST)

~ Path 4 indicates that some grassroots have carried out some digital exploration under the combination of certain cognitive environments and normative environments, but there is no good emotional environment or behavioral environment, and the environmental effect of policy implementation of grassroots digital construction has not played a positive role in helping. In other words, although the policy implementers have a clear cognition that digitization at the grassroots level is imperative, if their sense of experience in digitization is poor, and there are obvious problems of digital formalism and digital suspension in the behavioral

**Table 17. Cases and main views corresponding to Path 3.**

| Interview Cases | Interviewee | Main point |
|---|---|---|
| NK | 14 sheets ** | Digital duplication exists in large numbers, the coordination of digital products is not enough, digital personnel has not played its due role, and grassroots personnel have not experienced digital effectiveness. |
| NL | 15 Zang ** | With the combination of online and offline supervision, the degree of digital supervision is less than 50%. It is recommended to promote progress in and the level of digital supervision; due to the lack of upper law, the speed of government reform cannot keep up with the speed of company change, and as well as digital abuse digital duplication, digital formalism exists to a high degree, and can it be embedded into a network management through the way of digital business embedding, repeated digital assessment. |

**Table 18. Cases and main views corresponding to Path 4.**

| Interview Cases | Interviewee | Main point |
|---|---|---|
| NP | 19 Lee * | The low level of digitization in townships, lack of attention from leaders, and insufficient manpower are the main limiting factors; the village community management team has a serious aging population and it does not have the quality and level of digital governance, the sustainability of digital governance policies is poor, and the phenomenon of task-based governance exists. |
| Na | 30 Liu ** | Digital personnel exist to make up numbers in order to determine a good cadre line; how can digital governance overcome digital formalism? The root cause is the contradiction between the standardization, standardization, and refinement of digital governance and the complexity, uncertainty, and roughness of grassroots governance, which leads to the problems of digital floating and digital formalism in the process of digitization at the grassroots level. There is still a long way to go to overcome digital floating and digital formalism. In recent years, the digital integration and coordination of the tax system have been greatly improved, but because of the professionalism and complexity of the tax itself, it is more difficult to use the service object. Therefore, the future needs the two-way interaction and efforts of the tax bureau and taxpayers to promote the improvement of the efficiency of tax governance at the grassroots level. |

environment, the policy incentive role of the overall policy implementation environment will be seriously restricted. The interview cases corresponding to this path are NP and Na, that is, interviewees "19 Li *" and interviewees "30 Liu **". Their main views are shown in Table 18. From their views, it can be seen that they do not have a good digital experience of their grassroots units, and there are a large number of digital floating and digital formalism problems in the policy implementation environment where they live. All these have restricted the process of digital construction of their grassroots units. This path indicates that in the cognitive and normative environment, grassroots organizations should pay attention to the cultivation of a good sense of experience for policy implementation subjects and the optimization of their behavioral environment and strengthen the governance of digital alienation phenomena such as digital formalism and digital suspension.

~ Path 5: (cognitive dimension) *(~ behavioral dimension) *(normative dimension) *(~ control dimension) (CO * be * ST * ct)

~ Path 5 indicates that in the grassroots digitization construction, even if certain benign cognitive and normative dimensions are generated, but no positive behavioral and control dimensions are formed, such a policy implementation environment cannot have a positive intervention effect on grassroots digitization. The interview cases corresponding to this path are Na and Nd, that is, interviewees "19 Li *" and interviewees "30 Liu **". Their main views are shown in Table 19. From their views, it can be seen that some grassroots organizations with better cognitive and normative environments are subject to the complexity, uncertainty, and roughness of the grassroots. It is difficult to give full play to the advantages of digital standardization, standardization, and fine governance, which means that the grassroots digital behavior environment is full of digital formalism, digital suspension, digital fragmentation and other phenomena. Despite the construction of digital control environment by digital means, there is a phenomenon of digital control fraud, which seriously restricts the positive role of the policy implementation environment in the digital process. This path indicates that for the grassroots with the above environmental characteristics, efforts should be made to remediate the behavioral environment and control environment in their digital construction, especially the problems of digital formalism and digital control fraud.

~ Path 6: (cognitive dimension)* (emotional dimension)* (~ behavioral dimension)* (normative dimension)* (control dimension) (CO * EM * be * st * CT)

**Table 19. Cases and main views corresponding to Path 5.**

| Interview Cases | Interviewee | Main point |
|---|---|---|
| Na | 30 Liu ** | Digital personnel exist to make up numbers in order to determine a good cadre line; how can digital governance overcome digital formalism? The root cause is the contradiction between the standardization, standardization, and refinement of digital governance and the complexity, uncertainty, and roughness of grassroots governance, which leads to the problems of digital floating and digital formalism in the process of digitization at the grassroots level. There is still a long way to go to overcome digital floating and digital formalism. In recent years, the digital integration and coordination of the tax system have been greatly improved, but because of the professionalism and complexity of the tax itself, it is more difficult to use the service object. Therefore, the future needs the two-way interaction and efforts of the tax bureau and taxpayers to promote the improvement of the efficiency of tax governance at the grassroots level. |
| Nd | 33 holes ** | Optimistic about the trend of digitalization, that it is inevitable, the current restriction of digitalization is mainly the cognition and quality of grassroots staff, management inertia and fragmentation of digital products. |

~ Path 6 indicates that a good policy behavior environment has not been generated in the grassroots digital construction. Even if the other four kinds of environments are generated, the positive supporting role of the policy implementation environment cannot be well played. Therefore, no matter how good the policy is, it should be attributed to implementation, and the policy without implementation will become a castle in the sky. The interview case corresponding to the path is NN, that is, the interviewer "37 Li **". She said in the interview: "The high level of digitalization of the public security department, so that people over 70 years old to adapt to the digitalization is difficult, data barriers exist, regional gap is large limit data interconnection, data security and personal privacy protection affect data integration." This path shows that in the grassroots units with early digitization and a relatively high digitization level, digital barriers have not been broken through, which is the main constraint factor for the implementation of the digital policy of the grassroots organization. In the future governance, digital coordination and integration should be paid attention to.

## 5. Conclusion and prospects

### 5.1 Conclusion

This study mainly consists of two parts: First, using NVivo12plus software and rooted theory for reference, the responses from interviews with 40 grassroots civil servants is analyzed via rooted coding. Then, based on the clear set qualitative comparative analysis method and rooted coding using the software Tosmana, a comparative analysis of the effect of the policy implementation environment is carried out for grassroots digital construction in China.

The research results of rooted coding show that:

1. The cognitive dimension is the logical starting point for the generation of the policy implementation environment in China's grassroots digital construction. Under the dual influence of self-cognition and external cognition, the cognitive environment plays a leading role in the generation of policy implementation environment in grassroots digital construction. The cognitive environment dimension consists of internal cognition and external cognition.

2. The emotional dimension is the comprehensive embodiment of the subject's inner attitude towards external things. It is composed of a variety of feelings. It is a derivative of cognition. It constitutes the emotional link generated by the policy implementation environment in

China's grassroots digital construction. The emotional dimension is composed of a sense of security, a sense of experience and a sense of boundary.

3. The behavioral dimension is the sum of the activities produced by the subject under the influence of various internal and external stimuli. It is the active response made by the subject under specific conditions. It is a key link in the formation of the policy implementation environment in China's grassroots digital construction as it represents the activity and process by which the policy implementation subject implements the policy under the influence of the cognitive and emotional environments. It is an important reference link to regulate and control the environment of peacekeeping. Its actual process is a test of whether the policy implementation environment is conducive to policy implementation, and it will also have an impact on the emotional environment of cognition and peacekeeping. It plays an important role in the whole process of generating the policy implementation environment in China's grassroots digital construction. It is the most active, uncertain and complex key link in the generation of the whole policy implementation environment.

4. The normative dimension refers to the standards that affect the subject. It consists of the formal legal system and informal ideology, values, cultural customs and so on. It has an incentive and restraint effect both inside and outside the main body. It is another key link in the generation of policy implementation environment in the digital construction of China. It provides not only important criteria and guidance for the generation of the behavioral dimension but also fertile soil for the integration of the cognitive, emotional and behavioral dimensions. On the one hand, the policy implementation norm dimension in China's grassroots digital construction is the focus and internalization of the entire policy environment at the implementation level. On the other hand, it is restricted and affected by the overall policy environment; thus, the two interact with each other.

5. Control environment maintenance refers to the activities and processes in which the subject is restricted to a certain extent by internal and external conditions. It limits the subject's activity to within a certain threshold. It is the correction and perfection of the subject's behavior. It is the continuation link of the standard environment and the behavior environment. It is also an important link in the generation of the policy implementation environment in China's grassroots digital construction. It is the inflection point of the closed-loop management of the policy implementation environment in China's grassroots digital construction within a certain policy time limit. Therefore, it is not only the end point but also the starting point, or the link that is associated with the cognitive, emotional, behavioral and normative environment dimensions, as well as every other link within the policy environment, at any given time.

The results of the clear set qualitative comparative analysis show the following:

1. There are six combination paths to produce the environmental effect of policy implementation in China's grassroots digital construction: Path 1: (cognitive dimension) * (~ emotional dimension) * (behavioral dimension) * (normative dimension); Path 2: (cognitive dimension) * (~ behavioral dimension) * (~ normative dimension) * (control dimension); Path 3: (cognitive dimension) * (~ affective dimension) * (behavioral dimension) * (control dimension); Path 4: (~ cognitive dimension) * (affective dimension) * (~ behavioral dimension) * (~ normative dimension) * (control dimension); Path 5: (cognitive dimension) * (affective dimension) * (~ behavioral dimension) * (normative dimension) * (control dimension); Path 6: (~ cognitive dimension) * (~ affective dimension) * (~ behavioral dimension) * (normative dimension) * (~ control dimension). In particular, Path 2, namely,

the combined mode wherein the cognitive, non-behavioral, non-normative and non-control environment dimensions are superimposed, covers the highest number of cases in policy implementation environment.

2. The cognitive environment dimension is the most critical link in the environmental governance of policy implementation at the initial stage of China's grassroots digital construction.

3. At present, the policy implementation environment in China's grassroots digital construction is dominated by the cognitive dimension, and the strong environmental effects of policy implementation have not yet been fully realized.

4. At present, the policy implementation environment in grassroots digital construction in China lacks strong mobilization of policy implementation subjects, and their enthusiasm and initiative cannot be fully harnessed.

5. At present, the policy implementation environment in grassroots digital construction is not responsive enough to the actual grassroots, fails to adequately adapt to the complexity and uncertainty of the grassroots contexts and, thus, cannot achieve grassroots digital construction.

6. At present, the policy implementation environment in grassroots digital construction encounters many digital problems, such as fragmentation, barriers, formalism, and abuse, which should be solved in the future.

7. At present, a grassroots digital policy implementation environment that considers the five dimensions that contribute to the environmental effects of policy implementation has not yet been generated. There is a long way to go before creating and developing a benign policy environment that integrates these five factors.

## 5.2 Countermeasures and suggestions

Through root theory tracing and clear set qualitative comparative analysis, this study draws the following policy suggestions to promote the generation of the policy implementation environment and the play of policy effects in China's digital construction:

**(1) Strengthen the top-level design and improve the degree of integration and coordination of the environment.** First, digitization at the grassroots level is a pain point, difficulty and key point in the construction of digital China. Grassroots units are very different, and grassroots affairs are numerous. The digital process is a process of high technology, which means it has a high threshold and high standard. Therefore, the digitization at the grassroots level is not a general grassroots policy the implementation subject can plan as a whole; whether it is a network management or a network office, it needs to strengthen the top-level design and draw a blueprint to the end. Second, digitization is a technological revolution. This revolution will involve changing the pattern of existing interests; many interests of the damaged may also hold a lot of traditional power, and the grassroots is the end of power operation. Its implementation environment is more difficult to supervise and control; therefore, we must rely on the top-level "sword" in order to reduce the implementation of grassroots policy discounted or out of the phenomenon. Third, grassroots digitization is still in the exploration stage of "crossing the river by feeling the stones". In the current grassroots digitization exploration, there is a lot of digital fragmentation, digital duplication, and digital abuse and numerous digital islanding problems, which need to be promoted from the top to the grassroots. To sum up, it is necessary to strengthen the top-level design and improve the degree of integration and coordination of

the environment, so as to promote the overall optimization of the policy implementation environment in China's grassroots digital construction.

**(2) Take root at the grassroots level and improve the adaptability of the environment.** From the interview survey, it can be clearly seen that the most reflected aspect of policy implementation behavior in China's grassroots digital construction is that existing digital products do not fit in with the reality of the grassroots, which means that digitalization not only fails to empower the grassroots but also becomes a digital burden for the grassroots. Therefore, in order to achieve high-quality top-level design, it is necessary to take root in the grassroots and frontline, so that the created policy implementation environment is in line with the grassroots digital reality, and truly help grassroots digitalization. On the one hand, this research involved meeting with the grassroots; collecting evidence, obstacles or ideas in the grassroots digital construction; fully considering the heterogeneity of the grassroots; and paving the way for the stratified pilot demonstration. A blueprint is not a one-size-fits-all but also needs to be combined with the actual grassroots, stratified echelons. On the other hand, grassroots digitization should pay attention to the stage; for the grassroots, the period of a stage has its own characteristics, the grassroots affairs cannot be digitized, the focus of each stage should be grasped, and there are key stages to adapt to the digitalization and avoid digital abuse, so as to improve the adaptability of the grassroots digital policy implementation environment.

**(3) Increase publicity and promotion to improve the experience of the environment.** At present, digital governance is still in the stage of hot and cold, the grassroots awareness of the digital-related policies of the superior is not yet in place, and many grassroots do not know how to implement them. On the one hand, we should strengthen the system construction, optimize the standard environment of policy implementation in the grassroots digital construction, solve the current legalization of many grassroots constraints on digital factors, and provide raw materials for the publicity and promotion of the policy environment. On the other hand, some incentive policies should be introduced to stimulate and guide the grassroots to actively use digital products through preferential policies, so that the subject and object of the grassroots digital policy service have the inherent demand to experience the digital governance method. For example, the grassroots government or grassroots policy implementation subjects use digital means to carry out public governance and services as an indicator of assessment and incentive, and give material and spiritual rewards to stimulate their active participation in grassroots digital governance and guide them to help grassroots service objects experience digital products and the convenience brought by digital products. In addition, it is necessary to pay attention to the cultivation and creation of digital culture, actively publicize and promote digital culture and concepts, and promote the deep integration of grassroots and digitalization, so that grassroots policy implementers can create a good experience brought by digitalization in the digital atmosphere.

## Acknowledgments

This research was supported by the annual project of Hebei Social Science Foundation "Research on the Development History of Hebei Coastal Cities" (No. HB18WH06) and the Research project of Humanities and Social Sciences of Hebei Colleges and Universities "Research on the Development of Hebei Province's Characteristic Industries towards the Sea" (No. ZD202406).

## Author Contributions

**Conceptualization:** Junjie Li, Bangfan Liu.

**Data curation:** Junjie Li.

**Formal analysis:** Junjie Li.

**Funding acquisition:** Junjie Li.

**Investigation:** Junjie Li.

**Methodology:** Junjie Li.

**Project administration:** Junjie Li.

**Software:** Junjie Li.

**Supervision:** Bangfan Liu.

**Validation:** Bangfan Liu.

**Visualization:** Bangfan Liu.

**Writing – original draft:** Junjie Li, Bangfan Liu.

**Writing – review & editing:** Bangfan Liu.

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
