## [Decision Letter · Decision Letter 0]

30 Jan 2024

PONE-D-23-22945Research on the generative logic and Configuration effects of the policy implementation environment in China's grass-roots digital construction: traceability based on grounded theory and validation of csQCA methodPLOS ONE

Dear Dr. Bangfan,

Thank you for submitting your manuscript to PLOS ONE. After careful consideration, we feel that it has merit but does not fully meet PLOS ONE’s publication criteria as it currently stands. Therefore, we invite you to submit a revised version of the manuscript that addresses the points raised during the review process.

We look forward to receiving your revised manuscript.

Kind regards,

Tinggui Chen

Academic Editor

PLOS ONE

Journal Requirements:

2. You indicated that ethical approval was not necessary for your study. We understand that the framework for ethical oversight requirements for studies of this type may differ depending on the setting and we would appreciate some further clarification regarding your research. Could you please provide further details on why your study is exempt from the need for approval and confirmation from your institutional review board or research ethics committee (e.g., in the form of a letter or email correspondence) that ethics review was not necessary for this study? Please include a copy of the correspondence as an ""Other"" file.

   " NO.The funders had no role in study design, data collection and analysis, decision to publish, or preparation of the manuscript."

Additional Editor Comments:

Thank you for submitting your manuscript to PLOE ONE.

I have completed my evaluation of your manuscript. The reviewers recommend reconsideration of your manuscript following major revision. I invite you to resubmit your manuscript after addressing the comments below.

Reviewers' comments:

Reviewer's Responses to Questions

**Comments to the Author**

1. Is the manuscript technically sound, and do the data support the conclusions?

Reviewer #1: Yes

Reviewer #2: Yes

2. Has the statistical analysis been performed appropriately and rigorously? 

Reviewer #1: Yes

Reviewer #2: Yes

3. Have the authors made all data underlying the findings in their manuscript fully available?

Reviewer #1: Yes

Reviewer #2: Yes

4. Is the manuscript presented in an intelligible fashion and written in standard English?

Reviewer #1: No

Reviewer #2: No

5. Review Comments to the Author

Reviewer #1: This manuscript Using NVivo12plus software, this paper constructs a generation model of policy implementation environment in China's grass-roots digital construction by taking 37 Chinese grass-roots civil servants' interview texts as the research object. Combining the results of the analyses, the authors make recommendations that can facilitate the creation of an environment for policy implementation and policy impact of digital construction in China. Overall, an interesting paper is presented on an important topic area. The writing style is well structured and logical and the methodology clearly delineated. An interesting and enjoyable read that deserves to be published subject to amendments.

The contributions and research questions are not mentioned clearly in the introduction section.

In the introduction section, authors need to state why the focus of the study is China.

Provide a reference for the claim stated in “...the policy implementation environment in grass-roots digitization construction involves many aspects such as organization, technology, system and mechanism, and is affected and restricted by compound causality, which is not suitable for linear causality explanation.”

In China, the Big Data Authority (BDA) is a key government body coordinating the construction and management of government information network systems, government data centres, e-government infrastructure, and basic and public government informatisation projects. So, in my opinion, the authors should explain why the research interview group did not include anyone from the BDA.

As the development of grass-roots digital government is highly correlated with local information infrastructure development and economic levels, variability is a factor that cannot be ignored. Is the research group covered by this manuscript representative as it focuses on a specific region (MPA students at University Yanshan)?

How to link the results with the Chinese governance practice is important for discussions especially when it involves the international implications.

The article has a lot of grammar mistakes, and the author should carefully check.

Other references are incorrectly cited and should be checked by the authors. When published in Chinese databases, the literature should not be directly translated for citation.

Fu Liping, Chen Qin, Dong Yongqing, et al. How does technological governance affect township cadres' actions? -- Based on the analysis of the implementation process of the targeted poverty alleviation policy in X City. Review of Public Administration, 21,14(04):119-136

Jiang Bao, Cao Taixin, Kang Wei. A study on the Organizational Structure Reform of Grass-roots Government driven by Digital government -- based on the case of Nanhai District Government of Foshan City. Journal of Public Administration, 202,19(02):72-81.

In order to facilitate the understanding and tracking of the papers by a global audience, authors should replace a large number of references from Chinese databases. The current way of citing references is fundamentally wrong.

Reviewer #2: 1. This is a good paper; however, it needs to be presented neatly in terms of formatting (spacing, line spacing). Tables must be justified.

2. The author needs to proofread this paper as the English is not good in terms of word choice and grammar. Example: Yuan Mingbao (2018) believes that digitalization and

textualization have brought about the lack of poverty governance. [1]Wang Yaling (2019) believes that the lack of public

participation leads to problems in smart city construction, such as focusing on technology over application, stereotyping,

insufficient public perception, and the digital divide between urban and rural areas. [2]Huang Jianwei et al. (2019) believe that

while it is important to solve the technical dilemma of grassroots digital governance, it is more important to pay attention to the

construction of ethics and ethics and promote the humanization of digital governance. Instead of using the word "believe," the author can opt for alternative terms to convey the same idea. For example: Yuan Mingbao (2018) argues that digitalization and textualization have resulted in the absence of effective poverty governance. [1] Wang Yaling (2019) asserts that the insufficient participation of the public leads to challenges in smart city construction, such as a disproportionate focus on technology over application, stereotyping, inadequate public perception, and a digital divide between urban and rural areas. [2] Huang Jianwei et al. (2019) posit that while addressing the technical challenges of grassroots digital governance is crucial, it is even more imperative to prioritize the construction of ethics and values, promoting the humanization of digital governance. Using terms like "argue," "assert," and "posit" adds a slightly more assertive tone to the statements while maintaining the author's attribution of perspectives to the respective researchers.

3. The author need to enhance the clarity and depth of the result and analysis: a) Define and Explain the Result Formula: Clearly define and explain the components of the result formula (CO * em * BE * ST). Provide a brief explanation of each variable to aid the reader's understanding. Example: Result Formula Explanation: The result formula, CO * em * BE * ST, represents the interaction of cognitive (CO), emotional (~ em), behavioral (BE), and normative (ST) dimensions. The "~" symbol denotes negation or the combination of lowercase letters. b) Connect Results to Overall Objective: Clearly articulate how Path 1 contributes to achieving the overall objective, emphasizing the positive environmental effects of policy execution in grassroots digital construction. Example: Path Contribution to Objective: Path 1 underscores the significance of generating cognitive, behavioral, and normative dimensions in policy execution within our country's grassroots digital construction. This path is crucial for realizing positive environmental effects stemming from effective policy implementation. c)Detailed Case Study Integration: Provide more detailed insights from the interview cases (NC, NX). Include direct quotes or specific examples to illustrate the perspectives of interviewees. Example: Interviewee Perspectives (Table 11): Table 11 outlines the main views of interviewees "03 Liu *" and "27 Zhang **." Their insights offer valuable nuances to the analysis, providing specific examples of challenges and opportunities within the identified path. d)Thorough Discussion of Obstacles: Elaborate on obstacles, such as leaders' digital cognition and quality. Discuss the practical implications and propose strategies for overcoming these obstacles. Example: Obstacle Analysis and Recommendations: The analysis reveals that leaders' digital cognition and quality present a substantial cognitive dimension obstacle. It is imperative to address the root cause— the relationship and transformation between digital and traditional organizational power. Future efforts should focus on straightening out this relationship to facilitate effective grassroots digital construction. e) Comparative Analysis Enhancement: Strengthen the comparative analysis between the digitization processes of the grassroots business side and the government side. Provide deeper insights into challenges and potential solutions. Example: Comparative Analysis of Digitization Processes: Despite considerable behavioral exploration by grassroots entities under state guidance, challenges persist. Notably, digital products lack grounding, digital fragmentation is prevalent, and organizational structures are not sufficiently flat. Moreover, a comparative analysis reveals that the digitization process on the grassroots business side outpaces that of the government side, emphasizing the need for targeted interventions. f) Future Recommendations: Offer specific recommendations for future actions, research, or policy changes based on the identified challenges and opportunities. Example: Future Recommendations: Based on the current analysis, future efforts should prioritize straightening out the relationship between digital and traditional organizational power. Additionally, a focused initiative to address challenges in digital product grounding, fragmentation, and organizational structure is warranted. Furthermore, creating a conducive emotional environment for grassroots digitalization should be a key consideration for policymakers and practitioners.

By incorporating these recommendations, the author can provide a more comprehensive and actionable analysis of the NVivo results.

4. In term of reference: a) Ensure consistency in how author names are presented. For instance, the first reference uses "Fang Fei, Zhang Enjian," while the fourth reference uses "Sun Zong-feng, Sun Yue." the author may consider standardizing the format. b)Ensure that the journal titles are accurate and consistently formatted. For instance, in the second reference, the journal title is "Leadership Science." Make sure this is the correct and complete title. c)Standardize the representation of page numbers. For instance, in the fourth reference, use "16-26" instead of "16(03):16-26" for clarity. d) If the publications have Digital Object Identifiers (DOIs) or URLs, consider including them for additional accessibility and verification.

6. PLOS authors have the option to publish the peer review history of their article (what does this mean?). If published, this will include your full peer review and any attached files.

Reviewer #1: No

Reviewer #2: **Yes: **Dr. Farah Adilla Ab Rahman

---

## [Author Response · Author response to Decision Letter 0]

10 Mar 2024

Dear review experts, thank you for reviewing our manuscript. 

According to your review opinions, we have made a comprehensive revision. The contents of the revision mainly include: first, the introduction part of the manuscript has been comprehensively revised, especially for the revision opinions of the review experts; Second, the second part of the manuscript has been modified, focusing on the source selection of literature, data collection and big data management issues, and adding a paragraph of discussion; The third is to make targeted changes to the table, diagram, formatting and participating documents according to the standard of this journal; Fourthly, we hired MDPI professionals to make native language modifications to the use of language in our manuscript, so that our manuscript could achieve native language discussion.

For details of the changes, see the marked document. Please review it.

Salute to you!

Liu bangfan

2024/2/21

---

## [Decision Letter · Decision Letter 1]

20 Mar 2024

Research on the generative logic and Configuration effects of the policy implementation environment in China's grass-roots digital construction: traceability based on grounded theory and validation of csQCA method

PONE-D-23-22945R1

Dear Dr. Bangfan,

We’re pleased to inform you that your manuscript has been judged scientifically suitable for publication and will be formally accepted for publication once it meets all outstanding technical requirements.

Kind regards,

Tinggui Chen

Academic Editor

PLOS ONE

Additional Editor Comments (optional):

Reviewers' comments:

Reviewer's Responses to Questions

**Comments to the Author**

1. If the authors have adequately addressed your comments raised in a previous round of review and you feel that this manuscript is now acceptable for publication, you may indicate that here to bypass the “Comments to the Author” section, enter your conflict of interest statement in the “Confidential to Editor” section, and submit your "Accept" recommendation.

Reviewer #1: All comments have been addressed

Reviewer #2: All comments have been addressed

2. Is the manuscript technically sound, and do the data support the conclusions?

Reviewer #1: Yes

Reviewer #2: Yes

3. Has the statistical analysis been performed appropriately and rigorously? 

Reviewer #1: Yes

Reviewer #2: Yes

4. Have the authors made all data underlying the findings in their manuscript fully available?

Reviewer #1: Yes

Reviewer #2: Yes

5. Is the manuscript presented in an intelligible fashion and written in standard English?

Reviewer #1: Yes

Reviewer #2: Yes

6. Review Comments to the Author

Reviewer #1: As a result of the author's revisions, I think the quality of this manuscript has been greatly improved and is ready to be published.

Reviewer #2: It's evident that you've put in considerable effort to address all the necessary corrections. Wishing you all the best in your future endeavors and looking forward to more of your articles

7. PLOS authors have the option to publish the peer review history of their article (what does this mean?). If published, this will include your full peer review and any attached files.

Reviewer #1: **Yes: **Hongxin Ma

Reviewer #2: No

---

## [Editor Report · Acceptance letter]

26 Mar 2024

PONE-D-23-22945R1 

PLOS ONE

Dear Dr. Liu, 

I'm pleased to inform you that your manuscript has been deemed suitable for publication in PLOS ONE. Congratulations! Your manuscript is now being handed over to our production team.

Kind regards, 

on behalf of

Dr. Tinggui Chen 

Academic Editor

PLOS ONE